# The RNA-binding protein vigilin regulates VLDL secretion through modulation of *Apob* mRNA translation

Mehrpouya B. Mobin[1], Stefanie Gerstberger[2], Daniel Teupser[3], Benedetta Campana[4], Klaus Charisse[5], Markus H. Heim[4], Muthiah Manoharan[5], Thomas Tuschl[2] & Markus Stoffel[1]

The liver is essential for the synthesis of plasma proteins and integration of lipid metabolism. While the role of transcriptional networks in these processes is increasingly understood, less is known about post-transcriptional control of gene expression by RNA-binding proteins (RBPs). Here, we show that the RBP vigilin is upregulated in livers of obese mice and in patients with fatty liver disease. By using *in vivo*, biochemical and genomic approaches, we demonstrate that vigilin controls very-low-density lipoprotein (VLDL) secretion through the modulation of apolipoproteinB/*Apob* mRNA translation. Crosslinking studies reveal that vigilin binds to CU-rich regions in the mRNA coding sequence of *Apob* and other proatherogenic secreted proteins, including apolipoproteinC-III/*Apoc3* and fibronectin/*Fn1*. Consequently, hepatic vigilin knockdown decreases VLDL/low-density lipoprotein (LDL) levels and formation of atherosclerotic plaques in $Ldlr^{-/-}$ mice. These studies uncover a role for vigilin as a key regulator of hepatic *Apob* translation and demonstrate the therapeutic potential of inhibiting vigilin for cardiovascular diseases.

[1] Institute of Molecular Health Sciences, ETH Zurich, Otto-Stern Weg 7, 8093 Zurich, Switzerland. [2] Howard Hughes Medical Institute, The Rockefeller University, 1230 York Avenue, New York, New York 10065, USA. [3] Institute of Laboratory Medicine, Ludwig-Maximilians-University Munich, Marchioninistr. 15, 81377 Munich, Germany. [4] Department of Biomedicine and Clinic for Gastroenterology and Hepatology, University Hospital Basel, Hebelstrasse 20, 4031 Basel, Switzerland. [5] Alnylam Pharmaceuticals, 300 Third Street, Cambridge, Massachusetts 02142, USA. Correspondence and requests for materials should be addressed to M.S. (email: stoffel@biol.ethz.ch).

Vigilin (official gene symbol *Hdlpb*) is the largest RNA-binding protein of the KH domain family. It is conserved from human to yeast and has a unique structure of 14 imperfect tandem repeats of ∼70 amino acids that contain KH domains, sequence motifs that are usually found in RNA-binding proteins[1–3]. Some KH domains in vigilin have been shown to have nucleotide-binding activities[4], and vigilin has been found to be associated with cytoplasmic mRNA[5] and tRNA[6]. Vigilin has been implicated in diverse biological processes such as sterol metabolism[5,7–10], carcinogenesis[11,12], control of translation[13,14], formation of heterochromatin[15–17], nuclear export of tRNA[18], cytoplasmic transport of RNA[19] and metabolism of specific mRNAs[7,20–22]. Yet, very few mRNA targets and no precise RNA recognition element (RRE) have been reported in mammals. Furthermore, its function in mice and human has not been addressed in a systematic and unbiased manner.

Here, through unbiased approaches using photoactivatable ribonucleoside-enhanced crosslinking and immunoprecipitation (PAR-CLIP), RNA sequencing, as well as label-free quantification by mass spectrometry, we identify vigilin as a translational regulator of a subset of genes encoding for secreted liver proteins. We report that vigilin binds to CU-rich regions in the mRNA coding sequence of *Apob* and other proatherogenic secreted proteins, including apolipoproteinC-III/*Apoc3* and fibronectin/*Fn1*. Expression of vigilin also correlated with lipid accumulation in patients with non-alcoholic fatty liver disease (NAFLD) and non-alcoholic steatohepatitis (NASH) as well as livers of insulin-resistant obese mice. Lastly, we show that silencing of vigilin in the liver reduces atherosclerotic plaque formation in *Ldlr*$^{-/-}$ mice, suggesting a critical role of vigilin in hepatic metabolism and a possible therapeutic approach for the prevention of cardiovascular diseases.

## Results

**Vigilin regulates hepatic lipid metabolism.** Vigilin is ubiquitously expressed with highest levels in organs with preferential endodermal cell origin, including the liver (Fig. 1a), and is predominantly localized to the cytoplasm in hepatocytes (Fig. 1b). To assess its relevance in metabolic disorders, we investigated if vigilin was deregulated in obese, insulin-resistant mice and measured its expression in livers of diet-induced obese C57Bl/6J (DIO) as well as *ob/ob* mice (Supplementary Fig. 1a–c). While vigilin mRNA levels were similar (Supplementary Fig. 1d), hepatic protein levels of vigilin, but not other RNA-binding proteins (RBPs) with moderate and high expression levels such as HuR/*Elavl1* and Rbm47, were markedly increased in obese mice compared with chow-fed control animals (Fig. 1c). Furthermore, plasma triglycerides, non-esterified fatty acids (NEFA) and cholesterol correlated significantly with hepatic vigilin levels (Fig. 1d–f). We also compared vigilin protein levels in liver biopsies of 5 healthy, 10 NAFLD and 10 NASH patients (Supplementary Table 1) and found a strong positive correlation between the subjects' vigilin levels and their degree of liver steatosis (Fig. 1f,g).

To investigate the functional consequence of elevated vigilin expression in the livers of insulin-resistant obese mice, we generated a recombinant adenovirus expressing human VIGILIN (Ad-VIGILIN) that was administered intravenously to male C57Bl/6J mice via tail vein injections. This resulted in a liver-specific overexpression (Supplementary Fig. 2a), which was comparable to the ≈threefold increase of vigilin protein in *ob/ob* mice (Fig. 2a). Adenovirus-mediated overexpression of vigilin was sufficient to elevate plasma triglyceride and NEFA levels while lowering triglyceride content in the liver, when compared with Ad-GFP-infected, age- and weight-matched

control mice (Fig. 2b-d; Supplementary Fig. 2b). These changes in the lipid profile were also reflected by increased very-low-density lipoprotein (VLDL) levels as well as by oil red O stainings in the liver (Fig. 2e; Supplementary Fig. 2c). No changes in plasma glucose, insulin, cholesterol and alanine transaminase (ALT) were observed, indicating that increased hepatic vigilin expression does not affect glucose metabolism or cause hepatocellular injury (Fig. 2d; Supplementary Fig. 2d–f). These data show that vigilin expression is increased in steatotic livers of insulin-resistant subjects and that elevated hepatic vigilin levels increase triglyceride metabolism and VLDL secretion from the liver.

We next studied the effect of silencing vigilin by employing a recombinant adenovirus expressing an shRNA (Ad-shVig) that targets vigilin mRNA (*Hdlbp*). Injection of Ad-shVig resulted in a >90% reduction of hepatic vigilin in both chow and high-fat diet (DIO) C57Bl/6 mice when compared with phosphate-buffered saline (PBS) or control adenovirus injected mice expressing a nonfunctional shRNA (Ad-shCtrl; Fig. 2f). Knockdown of vigilin was restricted to the liver (Supplementary Fig. 2g) and both inflammation and hepatotoxicity markers *Nfkb*, *Tnfa*, *Il6* and ALT, respectively, remained in a normal range in adenovirus and PBS-injected mice (Supplementary Fig. 2h–j). Similar to the gain-of-function study, knockdown of hepatic vigilin did not affect glucose metabolism since no significant changes in blood glucose and insulin levels could be determined in chow or DIO mice under *ad libitum* fed conditions (Supplementary Fig. 2k–n). However in contrast to the hepatic vigilin overexpression, silencing of vigilin in chow diet mice resulted in increased liver triglyceride content and decreased plasma triglyceride as well as NEFA levels (Fig. 2g,h; Supplementary Fig. 2o–r). Knockdown of vigilin in DIO mice also lowered cholesterol levels through VLDL and low-density lipoprotein (LDL) particles when compared to Ad-shCtrl-injected control mice (Fig. 2i-j). In line with these observations, the correlations of vigilin expression levels with plasma triglyceride, NEFA and cholesterol levels were reversed upon knockdown of vigilin (Supplementary Fig. 2s).

**Genome-wide identification of vigilin targets.** To gain mechanistic insights into how vigilin influences lipid metabolism, we aimed to identify the direct targets of vigilin by performing PAR-CLIP in primary hepatocytes. PAR-CLIP takes advantage of T-to-C conversions that are created during reverse transcription as a result of incorporated 4-thiouridine (4SU) being covalently UV crosslinked to the RBP[23]. Hence, RNA sequences obtained from PAR-CLIP intrinsically contain the information of specific crosslinking events and distinguish true RNA–protein interactions from background RNA. Ad-VIGILIN-infected primary hepatocytes were cultured with 4SU before UV-irradiation to induce crosslinking of vigilin to the labelled cellular RNAs. Immunoprecipitation of the radiolabeled VIGILIN–RNA complex showed a distinct crosslink at ∼150 kDa (Fig. 3a). The RNA from this complex was converted into a cDNA library and analysed by deep sequencing. We used PARalyzer, an algorithm that calculates the density of T-to-C conversions in PAR-CLIP reads, to detect binding sites in two biological replicates[24]. Keeping only targets found in both replicates, vigilin crosslinked to 755 gene transcripts of which 744 were mRNAs, corresponding to ∼6% of the murine liver transcriptome (Supplementary Data 1). A total of 1,401 binding sites were identified, 1,165 of which were in mature mRNAs (Fig. 3a; Supplementary Data 2). Vigilin-binding sites predominantly resided in the coding sequence (CDS; 956) comprising RNA recognition elements of CHHC or CHYC sequence segments (H = A/C/U and Y = C/U; Fig. 3b–d). These PAR-CLIP binding sites were evenly distributed along the CDS of

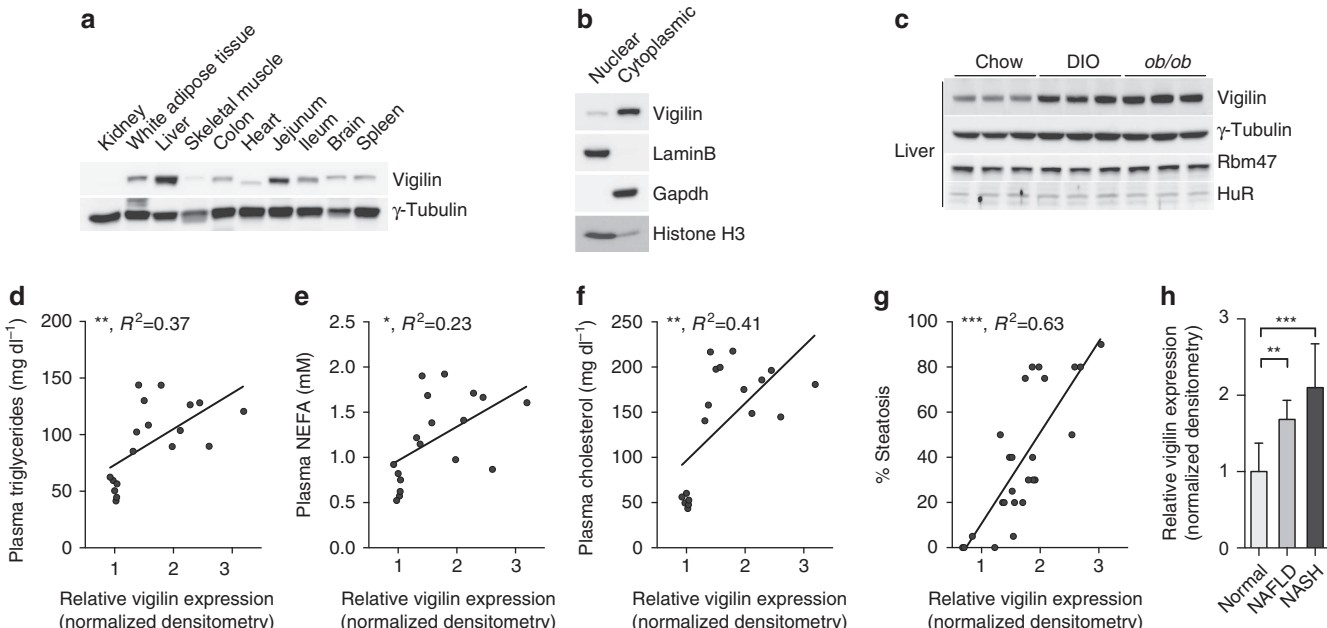

**Figure 1 | Correlation of vigilin levels with liver steatosis and plasma lipid levels.** (**a**) Western blot analysis of vigilin in mouse tissues. (**b**) Nuclear and cytoplasmic fractions from primary mouse hepatocytes. (**c**) Immunoblot analysis of hepatic vigilin levels in 20-week-old chow-fed (Chow) or high-fat diet C57BL/6 (DIO) and in 23 week old *ob/ob* mice. Correlation between hepatic vigilin expression quantified by densitometry from **c** (n = 6 per group) and (**d**) plasma triglyceride, (**e**) non-esterified fatty acid (NEFA) and (**f**) cholesterol levels. Correlation between (**g**) degree of hepatic steatosis and (**h**) clinical/histological diagnosis with hepatic vigilin expression quantified by immunoblotting and densitometry from human liver biopsies, including 5 healthy, 10 non-alcoholic fatty liver disease (NAFLD) and 10 non-alcoholic steatohepatitis (NASH) patients. Values in **h** are expressed as mean ± s.d. *$P \leq 0.05$, **$P \leq 0.01$, ***$P \leq 0.001$; P values and $R^2$ were determined by two-tailed Pearson's correlation test (in **d**–**g**) or ANOVA with Holm-Sidak *post hoc* analysis (in **h**).

targets (Supplementary Fig. 3a). Further *in silico* analysis of the motif revealed a preferred binding of vigilin to a tandem of CHHC or CHYC 4mers spaced by 2, 5 or 8 nt (Fig. 3e). We validated this putative RRE by electrophoretic mobility shift assays (EMSAs) using recombinant full-length human VIGILIN and synthetic single-strand RNAs comprising a panel of 18-nt di- and tri-nucleotide repeats. Consistent with the PAR-CLIP-derived RRE, RNA shifts were observed for CU-rich motifs, while insertions of A and G nucleotides resulted in decreased binding affinity (Fig. 3f; Supplementary Fig. 3b). Furthermore, when testing RNAs of different lengths we found that oligonucleotides of ≥18 nt were required to observe RNA–protein shifts, suggesting that, in addition to the RRE, RNA backbone contacts outside of the RRE are necessary for binding *in vitro* (Supplementary Fig. 3c). While its precise molecular function remains elusive, vigilin has been implicated in many cellular processes, including the stabilization and destabilization of specific mRNA targets[25,26]. To investigate vigilin's impact on its mRNA targets on a genome-wide scale, we profiled the transcriptome of livers by RNA sequencing upon overexpression and knockdown of vigilin. Strikingly, mRNA levels of the targets identified by PAR-CLIP did not change significantly under gain- or loss-of-function conditions (Fig. 3g,h), indicating that vigilin is not involved in direct stabilization or destabilization of its mRNA substrates.

Analysis of the PAR-CLIP data revealed that 99 of the top 100 target transcripts crosslinked with vigilin in the CDS and 78 harboured signal peptides, transmembrane domains or both, pointing towards a function for vigilin in the secretory pathway (Supplementary Fig. 4a). This notion is consistent with the reported role of the yeast homologue SCP160 (ref. 13), but distinct from that of other RBPs recently proposed to participate in translocation of alternative 3′ UTRs to the cell surface[27].

Overall, 393 of the ~2,250 liver-expressed secretory pathway proteins (~17.5%) were captured as targets with ≥4 T-to-C crosslinked reads in both PAR-CLIP replicates, indicating that only a specific subset of this class of proteins was targeted by vigilin. To assess if vigilin regulates the expression of secreted proteins on the protein level, we harvested the secretome from the medium of primary hepatocytes that were infected with either Ad-shCtrl or Ad-shVig and performed label-free quantification by mass spectrometry (MS-LFQ). We measured reduced levels of vigilin targets identified by PAR-CLIP in hepatocytes in which vigilin was silenced (Fig. 4a,b). Notably, the regulation of these proteins correlated with the number of crosslinked reads found in the PAR-CLIP analysis (Fig. 4c). The observed reduction of protein levels in MS-LFQ was validated for six metabolically relevant targets in the medium of Ad-shVig-treated primary hepatocytes by immunoblotting (Supplementary Fig. 4b). Five of them were also significantly regulated in the blood of mice upon adenovirus-mediated gain- and loss-of-function of vigilin (Fig. 4d). These targets included apoB, apoC-III, fetuin-A, alpha1-antitrypsin and orosomucoid with vigilin-binding sites evenly distributed along their CDS (Supplementary Fig. 4c). Since we used primary hepatocytes as an autonomous *ex vivo* system to study vigilin in the liver, the unchanged levels of fibronectin in the blood of mice with reduced hepatic vigilin expression is most likely due to compensatory secretion from extrahepatic tissues[28]. Importantly, the mRNA expression levels of these targets remained unperturbed upon overexpression or silencing of vigilin, thereby validating our genome-wide transcript analysis in hepatocytes with altered vigilin expression (Supplementary Fig. 4d,e; Fig. 3g,h).

**Vigilin as a translational enhancer of apoB.** Full-length apoB (apoB100) is a 550 kDa large core protein of VLDLs and LDLs

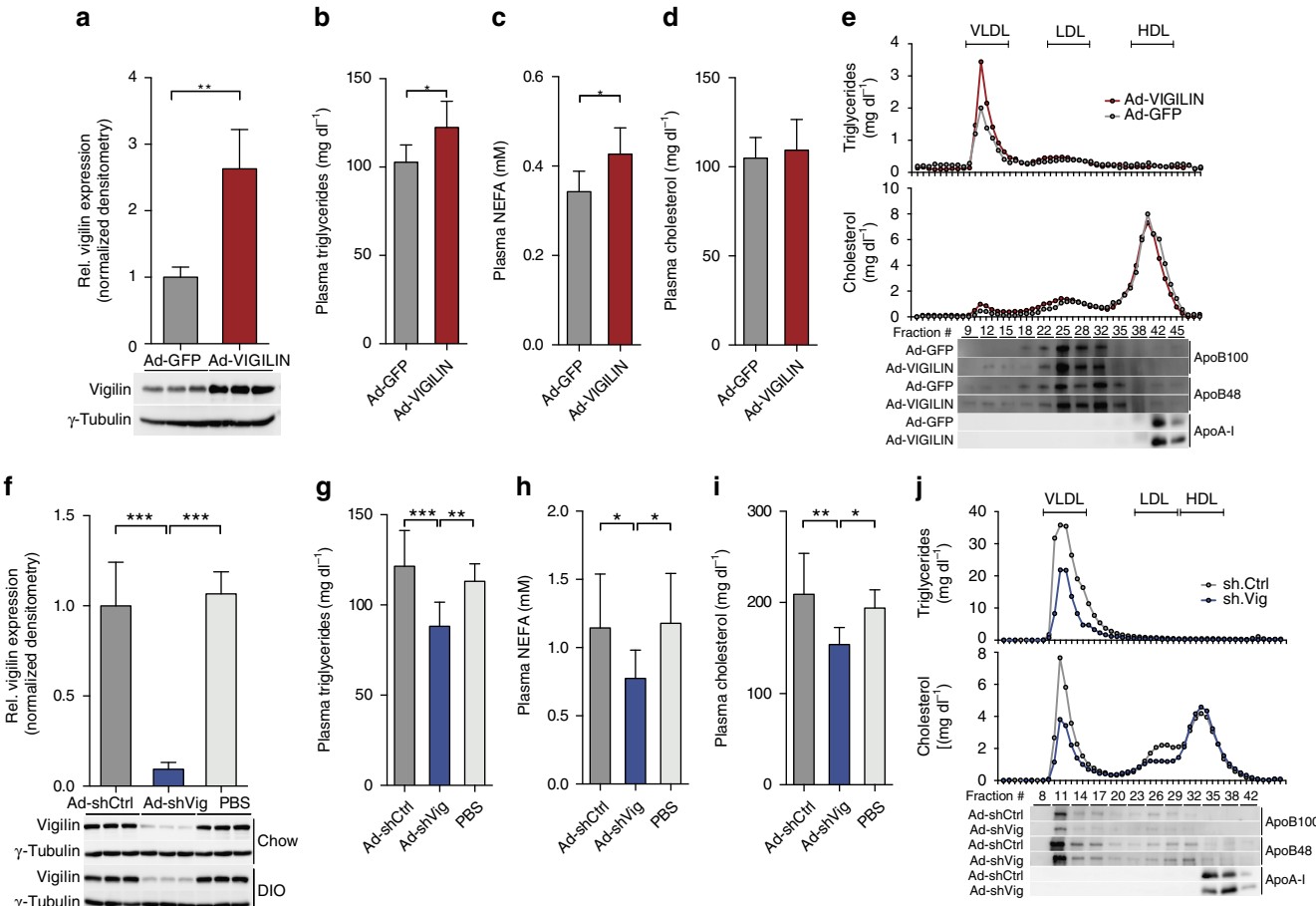

**Figure 2 | Hepatic modulation of vigilin regulates lipid metabolism.** (**a**) Relative expression of vigilin in livers of 8–10-week-old C57BL/6 mice injected with Ad-GFP or Ad-VIGILIN ($n = 5$ per group). Values are relative densitometric readouts normalized to γ-tubulin. (**b**) Plasma triglyceride, (**c**) NEFA and (**d**) cholesterol levels of mice from **a**. (**e**) Plasma from mice injected with Ad-GFP or Ad-VIGILIN was fractionated into very-low-density lipoprotein (VLDL), low density lipoprotein (LDL) and high density lipoprotein (HDL) particles and quantified through measurements of triglyceride and cholesterol levels in each fraction and used for western blot analysis of VLDL/LDL (apoB48/100) as well as HDL (apoA-I) markers. (**f**) Relative expression of vigilin in livers of 10-week-old chow-fed and 20-week-old high-fed diet C57BL/6 mice (DIO) with Ad-shCtrl ($n = 6$ for WT, $n = 8$ for DIO), Ad-shVig ($n = 6$ for WT, $n = 8$ for DIO) or PBS ($n = 3$ for WT, $n = 5$ for DIO). Values are relative densitometric readouts normalized to γ-tubulin. (**g**) Plasma triglyceride, (**h**) NEFA and (**i**) cholesterol levels of DIO mice treated as in **f**. (**j**) Plasma from DIO mice injected with Ad-shCtrl or Ad-shVig was fractionated into VLDL/LDL/HDL particles and quantified as in **e**. All values are expressed as mean ± s.d. *$P \leq 0.05$, **$P \leq 0.01$, ***$P \leq 0.001$; P values were determined by student's t-test (in **a**–**d**) or ANOVA with Holm-Sidak *post hoc* analysis (in **f**–**i**).

that are produced and secreted from the liver and supply peripheral tissues with triglycerides and cholesterol. With one copy per particle, apoB100 is an essential protein for the formation and secretion of VLDLs and LDLs. A second truncated isoform, apoB48, is the main apolipoprotein component of chylomicrons secreted from the intestine. In contrast to humans, murine apoB48 is also produced and secreted from hepatocytes. Increased plasma VLDL and LDL levels are independent risk factors for cardiovascular disease with further amplification when both VLDL and LDL cholesterol are elevated[29]. Fetuin-A is a small hepatokine (≈50 kDa) inhibiting ectopic calcification[30], but also influencing insulin[31–33] and inflammatory cytokine signalling in adipose tissue[30]. Notably, increase of both apoB and fetuin-A protein levels correlated with elevated expression of vigilin in DIO and *ob/ob* mice (Supplementary Fig. 4f). We employed apoB and fetuin-A as regulated proteins with strong PAR-CLIP signals as representatives for large and small transcript vigilin substrates, respectively, to study vigilin's interaction with their binding sites *in vitro*. To this end, we performed EMSAs to validate the affinity of vigilin to the respective PARalyzer determined mRNA binding sites identified by PAR-CLIP

(Fig. 4e). Whereas the wild-type sequences of respective binding regions induced a shift in electrophoretic mobility of recombinant, RNA-depleted VIGILIN, mutations of the identified RRE and scrambled sequences of these sites exhibited decreased or no binding affinity by human VIGILIN.

Given the reported association of the yeast[22] and vertebrate homologues[34] with ribosomes and our observation that vigilin positively regulates its targets post-transcriptionally, we hypothesized that vigilin influences protein production of a specific subset of proteins in the secretory pathway. We therefore investigated if vigilin was required for efficient translation of its targets by reconstructing translation *in vitro* using a cell-free system from liver extracts and fetuin-A as a model target followed by immunopurification of the translated protein. *In vitro* transcribed mRNAs of V5-tagged fetuin-A and apoM (as a non-vigilin-target) were efficiently translated and immuno-purified under wild-type conditions. In contrast, liver extracts with shRNA-mediated depletion of vigilin showed decreased translation of fetuin-A, but not apoM (Fig. 5a). In these extracts, fetuin-A production could be rescued by the addition of recombinant human VIGILIN. *In vitro* translation of apoB was

technically not feasible due to its ∼ 14 kb long mRNA. However, to provide further evidence that vigilin constitutes a positive regulator of translation for its targets, we performed metabolic labelling studies using [³⁵S]-methionine/cysteine to pulse primary hepatocytes that were isolated from Ad-VIGILIN or Ad-shVig-injected mice. Following immunoprecipitation of the radiolabeled

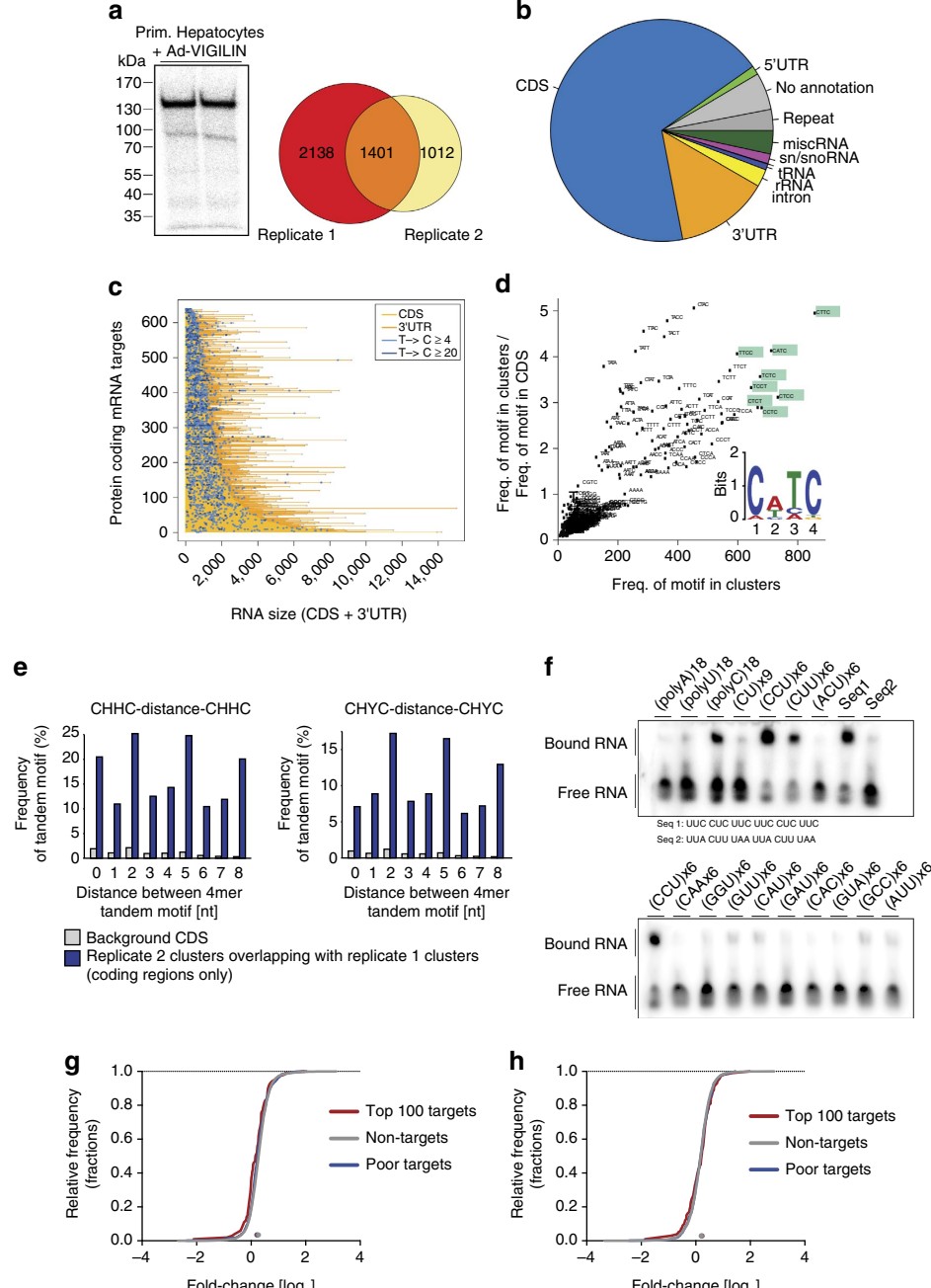

**Figure 3 | Vigilin binds to CU-rich sequences in the CDS.** (**a**) Autoradiograph of crosslinked, ³²P-labelled, VIGILIN–RNA immunoprecipitate separated by SDS–PAGE after PAR-CLIP and overlap of binding sites for the two biological replicates. (**b**) Distribution of PAR-CLIP binding sites (overlap of replicates) identified by PARalyzer in various RNA-species; (**c**) distribution of T-to-C reads along the CDS and 3′UTRs of mRNA targets. Bright (≥4 reads) and dark blue (≥20 reads) dots indicate positions of T-to-C reads along the transcripts. (**d**) kmer-plot and sequence logo representation of the vigilin RRE derived from PAR-CLIP binding sites: CHHC and CHYC (H = A/C/U; Y = C/U). (**e**) Enrichment analysis of vigilin PAR-CLIP clusters for tandem arranged RREs, previously derived from **d**, which are separated by 0–8 nt-long spacers. Background sequences are shuffled mouse CDS sequences. (**f**) Electrophoretic Mobility Shift Assays (EMSAs) validated affinity of vigilin to CU-rich sequences: synthetic RNAs representing 18-nt di- or tri-nucleotide repeats were radiolabeled (10 nM), incubated with 2 μM His₆-tagged recombinant human Vigilin and separated on 1% agarose gel. (**g,h**) Steady-state mRNA expression changes in livers of mice in which Vigilin was overexpressed using Ad-VIGILIN and compared with Ad-GFP (**g**) and livers of mice in which vigilin was silenced using Ad-shVig and compared to Ad-shCtrl (**h**) were determined by RNA-seq. Plotted is the empirical cumulative distribution function of vigilin PAR-CLIP targets (coloured lines) compared with expressed non-targets (FPKM ≥ 2, black and grey lines). Separately shown are the top 100 PAR-CLIP targets (based on cumulative crosslinked reads, red line), poor targets (remaining targets, blue line) and non-targets (not crosslinked in both PAR-CLIP replicates, grey line). The median transcript abundance change is indicated by a dot on the x axis.

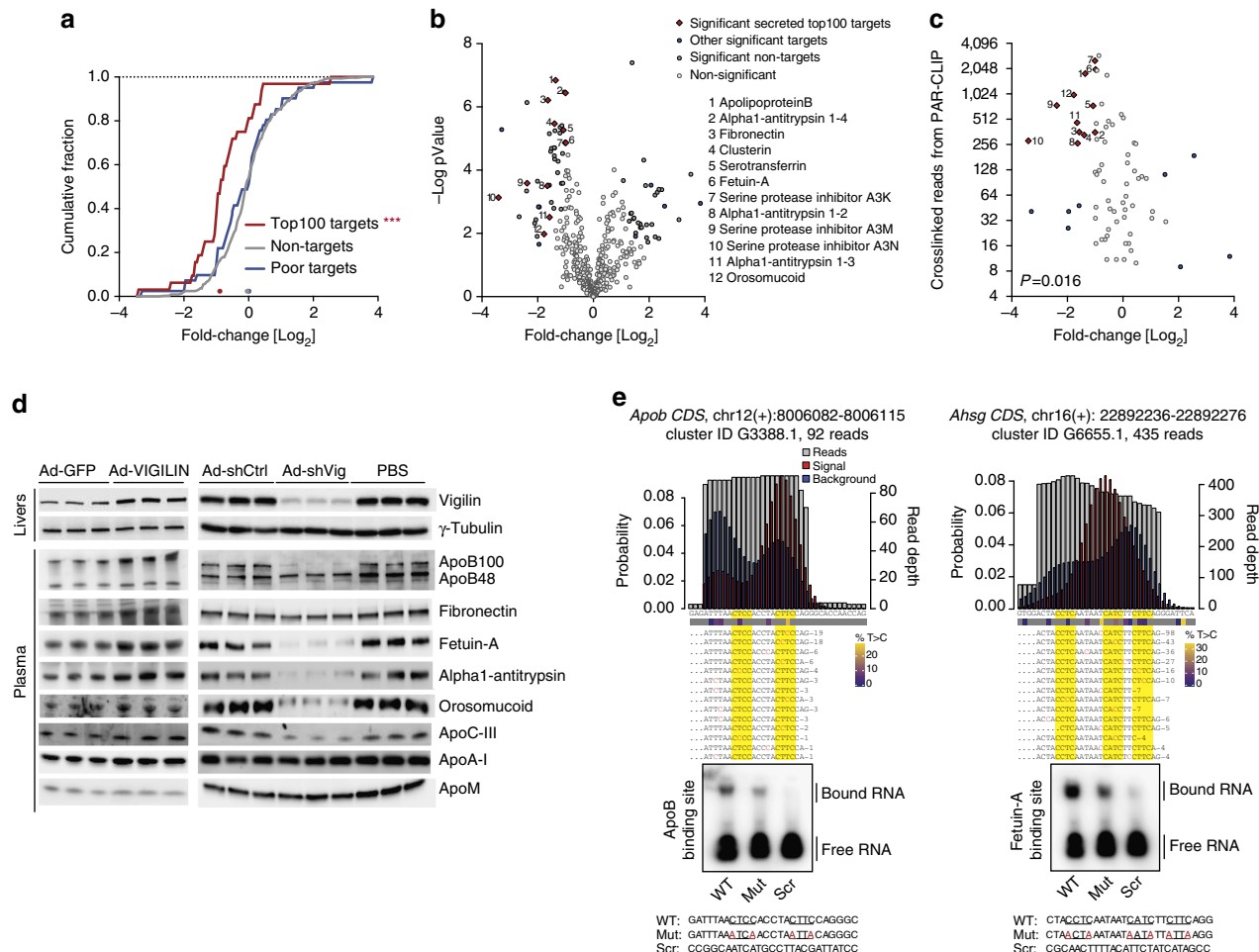

**Figure 4 | Vigilin controls levels of secretory proatherogenic proteins.** Secretome of primary hepatocytes isolated from 10-week-old mice injected with either Ad-shCtrl or Ad-shVig ($n=8$ per group; four biological replicates with each two technical replicates) was collected from the medium and quantified using label-free mass spectrometry (MS-LFQ). (**a**) Cumulative distribution function plot displaying fold-changes in secretion upon knockdown of vigilin in primary hepatocytes of top 100 PAR-CLIP targets (based on cumulative crosslinked reads, red line), poor targets (remaining targets, blue line) and non-targets (not crosslinked in both PAR-CLIP replicates, grey line). (**b**) Volcano plot of differentially secreted proteins upon vigilin knockdown in primary hepatocytes. x axis: $\text{Log}_2$ fold-change of intensities, y axis: $-\text{Log}_{10}$ P values. Significant hits among secreted top 100 PAR-CLIP targets (based on T-to-C counts) are indicated in red dots, other significant targets in blue, non-targets and non-significant hits in grey. Significance was determined using false discovery rate (FDR)-corrected (FDR = 0.01), permutation-based multiple $t$-tests ($250\times$) and curve bend s0 = 0.5. (**c**) Plot of differentially secreted PAR-CLIP targets (x axis) against T-to-C reads (y axis) indicates downregulation of more frequently bound targets. Significance was determined by Pearson's correlation test. (**d**) In vivo validation of MS-LFQ data through side-by-side immunoblot analysis of six targets from blood plasma of mice upon gain- (left panel: Ad-GFP and Ad-VIGILIN) and loss-of-function (Ad-shCtrl, Ad-shVig and PBS) from Fig. 2. (**e**) VIGILIN EMSAs representing binding sites on apoB and fetuin-A mRNAs identified by PAR-CLIP. Upper panel: alignment of vigilin PAR-CLIP sequence reads to gene loci of Apob and Ahsg (fetuin-A) mRNA CDS'. RREs are highlighted in yellow. The kernel density of T-to-C (T>C) transitions detected in PAR-CLIP reads is shown in red bars, the T-to-C conversion probability density of the cluster sequence is shown in blue bars. The read depth of the cluster is shown in grey. The percentage change of T-to-C transitions is indicated below the nucleotide sequence on a colour scale from blue to yellow. Lower panel: autoradiograph of EMSAs performed using binding site sequences identified by PAR-CLIP, mutated RREs (indicated in red) and scrambled sequences of these sites. The RNA sequences are indicated below.

targets, the effect of vigilin on target protein synthesis was quantified via scintillation counting. Consistent with the regulation of protein amounts in vivo, overexpression of vigilin led to increased target protein synthesis, while knockdown of vigilin resulted in decreased protein synthesis (Fig. 5b,c). We also investigated if vigilin influences post-translational degradation pathways that are known to determine the rate of secretion of apoB from the liver. To test if vigilin caused the observed regulation of apoB and fetuin-A by influencing proteolysis, we monitored half-lives of these proteins upon knockdown of vigilin. Supporting vigilin's primary role in protein synthesis, half-lives of both apoB and fetuin-A remained unchanged in primary hepatocytes upon cycloheximide-mediated translational

inhibition (Supplementary Fig. 5a,b). Together, these data support the role of vigilin as a translational enhancer and illustrate that protein synthesis of vigilin targets is directly affected by the amount of vigilin protein present during translation.

Given vigilin's impact on lipid metabolism and apoB as a main target, we next sought to confirm that vigilin is a key regulator of lipid secretion in hepatocytes. To this end, we pulse-chased primary hepatocytes with $^{14}\text{C}$-palmitate and measured secretion of thereby radiolabeled triglycerides into the medium. While $^{14}\text{C}$ counts were significantly higher in the medium of primary hepatocytes upon overexpression, less $^{14}\text{C}$ was detected upon knockdown of vigilin (Fig. 5d). The effect of vigilin on hepatic

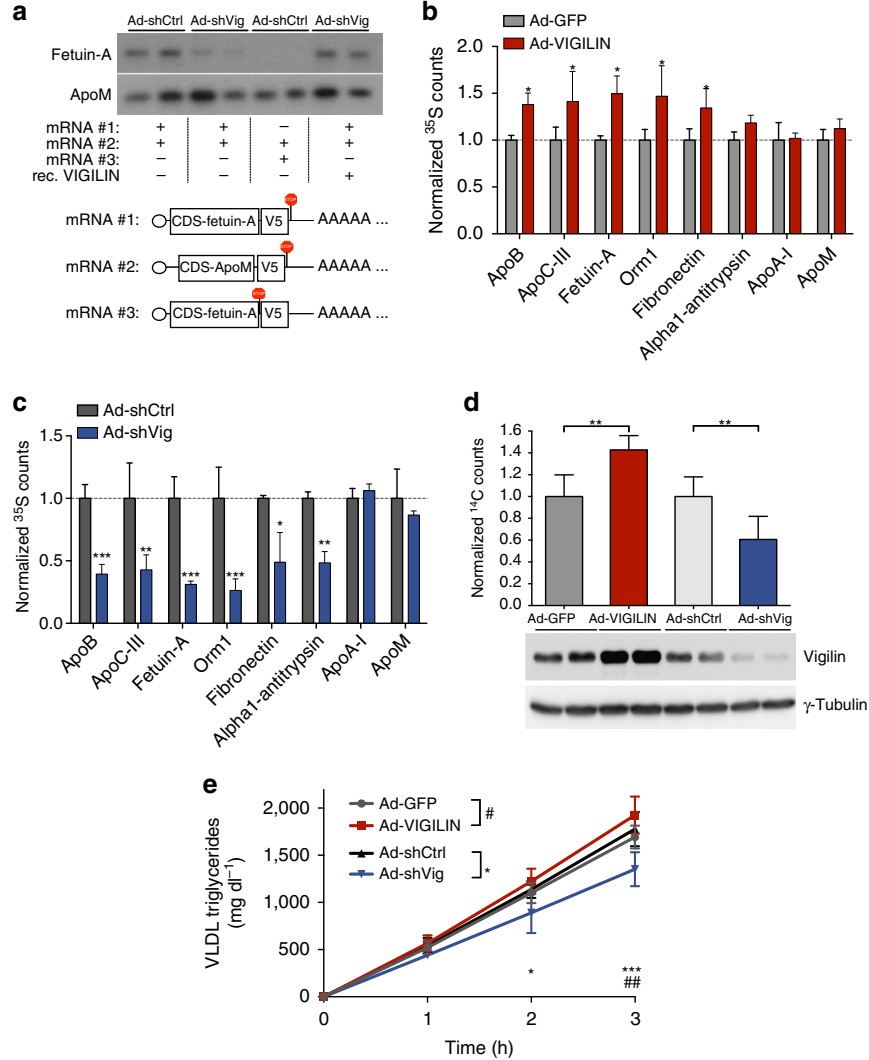

**Figure 5 | Vigilin enhances translation of its mRNA targets and lipid secretion.** (**a**) Autoradiograph of *in vitro* translation assays using fresh liver extracts from Ad-shCtrl or Ad-shVig-injected mice (*n* = 2 per group). Synthetic mRNAs (scheme indicated in lower panel) of fetuin-A and apoM (as control for non-target) were translated into V5-tagged and [35S]-methionine/cysteine radiolabelled protein, immunoprecipitated and separated by SDS–PAGE. Fetuin-A mRNA with premature stop-codon before C-terminal V5-tag (#3) was used as a negative control for immunoprecipitation. (**b,c**) 35S counts from metabolic labeling and immunoprecipitation of hepatic vigilin targets upon (**b**) overexpression (**c**) silencing of vigilin. Primary hepatocytes from mice injected with either Ad-VIGILIN (control: Ad-GFP) or Ad-shVig (control: Ad-shCtrl) were pulse-chased with [35S]-methionine/cysteine prior to immunoprecipitation of radiolabeled protein using target-specific antibodies and quantification *via* scintillation counting. Two biological and four technical replicates were used for each group. (**d**) 14C counts of radiolabeled palmitic acid incorporated into triglycerides and secreted into the medium by primary hepatocytes upon knockdown of vigilin. Primary hepatocytes were isolated from C57BL/6 mice injected with either Ad-GFP versus Ad-VIGILIN (for gain-of-function) or Ad-shCtrl versus Ad-shVig (for loss-of-function) and pulse-chased with 14C-labelled palmitic acid for incorporation into triglycerides. Lipids from the medium were extracted and quantified using 14C scintillation counting. Values are normalized relative counts in **b–d**. (**e**) VLDL triglyceride secretion assay in 8-week-old mice upon overexpression (Ad-VIGILIN; *n* = 6) or knockdown (Ad-shVig; *n* = 6) of hepatic vigilin protein. Animals received an intravenous injection of 500 mg kg$^{-1}$ tyloxapol to block lipases. Blood was collected at indicated time points and measured for plasma triglyceride accumulation. All values are expressed as mean ± s.d. *$P \leq 0.05$, **$P \leq 0.01$, ***$P \leq 0.001$; *P* values were determined by student's *t*-test (in **b–d**) or ANOVA with Tukey's *post hoc* analysis (in **e**).

lipid secretion was also assessed *in vivo* in a VLDL triglyceride secretion assay after blocking lipases with tyloxapol. Ad-shVig-injected mice exhibited a lower VLDL triglyceride secretion rate than their control littermates injected with Ad-shCtrl (Fig. 5e). Conversely, hepatic VLDL triglyceride secretion was increased upon overexpression of vigilin compared with control animals. Net triglyceride production controlled by vigilin was independent of altered triglyceride and VLDL clearance, since both lipoprotein lipase activity and VLDL uptake remained unchanged upon overexpression or silencing of vigilin (Supplementary Fig. 5c,d). Hence, these gain- and loss-of-function experiments demonstrate

that vigilin regulates lipid homoeostasis through the modulation of VLDL/LDL secretion by controlling the translation of apoB.

**Knockdown of vigilin in livers reduces atherosclerosis.** Short-term silencing of vigilin revealed decreased *de novo* expression of several proatherogenic proteins (apoB, apoC-III and fibronectin[35]), lowered VLDL, plasma triglyceride and NEFA levels. However, since vigilin targets many secreted proteins we could not rule out that the predicted anti-atherogenic effect of lowering hepatic vigilin is compensated by an unknown

combination of proatherogenic vigilin targets. We therefore evaluated the long-term effect of hepatic vigilin knockdown in atherosclerosis prone $Ldlr^{-/-}$ mice. Vigilin was silenced in the liver by two different siRNAs (siVig) that were chemically modified and covalently conjugated to multivalent N-acetylgalactosamine (GalNAc), a highly efficient ligand for clathrin-mediated endocytosis through the asialoglycoprotein receptor (ASGPR). We tested two siVig-GalNAc conjugates (GalNAc#1 and GalNAc#2) for activity. GalNAc#1 was less potent than adenovirus delivered shRNA (Ad-shVig) and weekly subcutaneous administration for 18 weeks resulted in a 70–80% reduction of hepatic vigilin. The second GalNAc-conjugated siRNA sequence (GalNAc#2) revealed no significant silencing activity and was used to control for chemistry-related effects (Fig. 6a). Knockdown of vigilin using GalNAc#1 was specific for the liver and showed no signs of liver toxicity (Supplementary Fig. 6a,b). We next measured the expression of vigilin targets in the plasma of siVig-GalNAc-treated mice by immunoblotting. The vigilin targets fetuin-A, alpha1-antitrypsin, orosomucoid and the proatherogenic apoB, apoC-III and fibronectin were decreased in siVig-GalNAc#1-treated mice compared with control animals with no changes in apoA-I and apoM (Fig. 6b). Plasma cholesterol and triglycerides were decreased in GalNAc#1 treated mice compared with PBS and GalNAc#2-injected control animals (Fig. 6c,d). Furthermore, plasma NEFA were reduced and lipid profiling of FPLC-separated lipoprotein fractions in plasma of these mice revealed decreased VLDL and LDL levels (Fig. 6e–g). Liver triglyceride and cholesterol content as well as plasma bile acids were similar in vigilin knockdown and control mice (Fig. 6h–j). Lastly, characterization of atherosclerosis in mice treated with siVig-GalNAc#1 revealed smaller lesions in H&E- and oil red O-stained aortic root sections (Fig. 6k,l). Together, these data demonstrate that long-term knockdown of vigilin in the liver decreases VLDL and LDL levels and reduces atherosclerotic plaque formation in mice.

## Discussion

Through the combination of in vivo gain- and loss-of-function studies, PAR-CLIP as well as label-free quantification of secreted proteins by mass spectrometry, the present study investigated the molecular and physiological role of vigilin in the liver. We identify vigilin as a positive regulator of translation for mRNAs coding for a subset of proteins of the secretory pathway by binding to an RRE consisting of a tandem repeat of CHHC separated by 2–8 nt. However, only a fraction ($\sim 17.5\%$) of all liver-expressed secretory pathway proteins were targeted by vigilin. Since the identified RRE is also present in other abundant mRNAs that are not bound by vigilin, the specific regulatory role of vigilin in protein translation may be dependent on additional protein interactions with the ribosome and other regulatory factors. Overall, these findings indicate a role of vigilin at the site of protein synthesis and are in good accordance with the recently proposed function of the yeast homologue SCP160 as a translational enhancer for proteins of the secretory pathway[13].

Our data revealed apoB among the strongest targets of hepatic vigilin. As the core protein of VLDL and LDL particles, apoB is of paramount importance for maintaining triglyceride balance within the liver. Although regulation of apoB by post-translational degradation pathways within the ER, post-ER[36], and by autophagy[37,38] is well established, translational regulatory mechanisms that govern apoB expression are poorly understood[39]. Our study demonstrates that vigilin is a major determinant of apoB translation and VLDL secretion by hepatocytes without affecting VLDL clearance and therefore a regulator of net triglyceride secretion by the liver.

Our current understanding of VLDL synthesis and regulation indicates that lipid availability constitutes the limiting factor for the synthesis of functional apoB100, and thus, for the assembly of secretion-competent VLDL within the ER lumen[40]. We would therefore expect a higher impact of vigilin on VLDL secretion in obese states, when lipids are abundant and less limiting. Hence, our results indicate that increased expression of vigilin enhances apoB synthesis in conditions of lipid availability (i.e., in obesity) to further promote VLDL secretion. Indeed we noticed a larger reduction of VLDL and apoB amounts upon knockdown of vigilin in DIO mice when compared with chow-fed animals (33% versus 51%, respectively), underlining the importance of lipid availability for apoB/VLDL secretion.

Short-term and strong ($>90\%$) silencing of vigilin (using Ad-shVig) in the liver was accompanied by mild steatosis, consistent with studies in which Apob was silenced using shRNAs[41]. Long-term knockdown of vigilin via GalNAc-conjugated siRNAs did not result in a steatotic liver. Protection against hepatic triglyceride accumulation is likely due to other vigilin targets and associated mechanisms, such as the reduction of apoC-III, which increases the catabolism of triglyceride rich particles and lowers plasma triglyceride levels in mice and humans[42–45]. Long-term silencing of hepatic vigilin in $Ldlr^{-/-}$ mice was also sufficient to lower VLDL and LDL levels and reduce atherosclerotic plaque formation. The anti-atherogenic effects resulting from the inhibition of vigilin expression are likely mediated mainly by apoB. However we cannot exclude systemic synergy of other vigilin targets such as apoC-III, fibronectin and possibly others. Fetuin-A has been reported to inhibit insulin and proinflammatory cytokine signalling. Yet as a mineral carrying protein, it also plays an important role in inhibiting systemic calcification and therefore might have facultative effects on atherosclerosis[46,47]. It will therefore be important to address possible synergism of vigilin targets by performing simultaneous knockdowns of these targets in the liver in relevant atherosclerosis models.

Knockdown of vigilin resulted in substantially decreased protein levels of its targets without affecting mRNA levels. In contrast, other highly expressed liver secreted proteins that were not identified as targets in both PAR-CLIP replicates such as fetuin-B or the lipoprotein apoE, showed no significant perturbation upon vigilin knockdown, further substantiating vigilin's specificity for a distinct subset of metabolic transcripts. Hence, our findings support the emerging view of RBPs organizing nascent RNA transcripts into functional groups that are coordinately regulated, especially at the level of mRNA stability and translation[48]. Combinatorial binding of additional RBPs to these mRNAs and their controlled proteasomal degradation may provide a mechanism for multi-dimensional regulation of protein fates and explain (1) the general poor correlation between the mRNA and protein pools in eukaryotic cells[49,50] and (2) the low susceptibility of some targets towards a vigilin knockdown ex and in vivo as observed for apoA-I. Targets that remained unchanged upon vigilin silencing may be subject to further regulatory mechanisms controlling their protein levels or be due to secondary effects of the resulting phenotype.

Taken together, we provide first evidence that the mammalian vigilin is involved in translational regulation of mRNA targets encoding for proteins of the secretory pathway, including the proatherogenic proteins apoB, apoC-III and fibronectin, and thereby serves as an important regulator of protein production and lipid secretion from the liver. Moreover, we uncover a primary role in lipid metabolism and hepatic steatosis in mice and humans to be mediated by an RNA-binding protein. The increased expression of vigilin in patients with hepatic steatosis may contribute to the overproduction and secretion of VLDL in

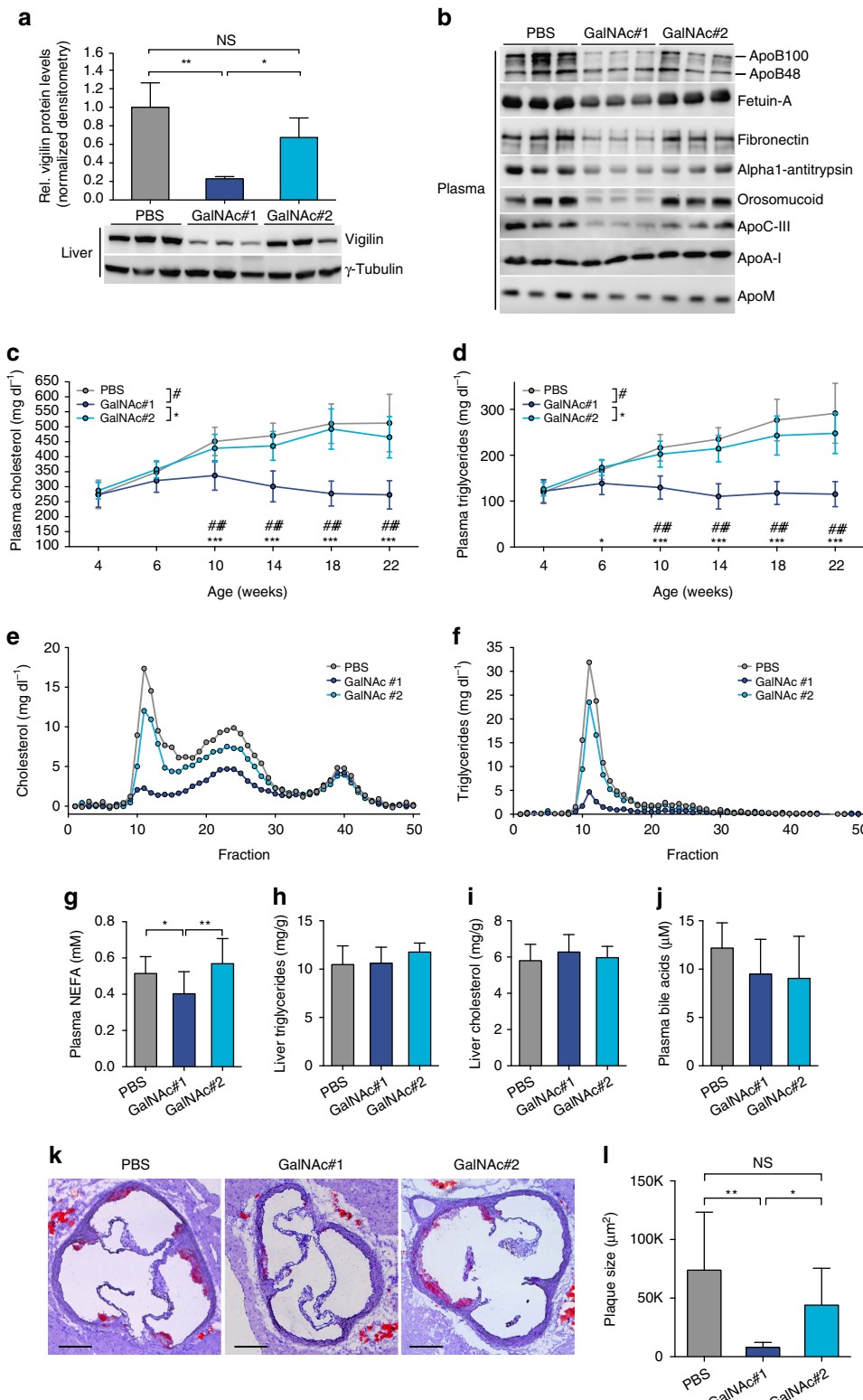

**Figure 6 | Knockdown of hepatic vigilin reduces atherosclerotic plaque formation.** (**a**) Quantification of hepatic vigilin knockdown in male $Ldlr^{-/-}$ mice with weekly injections of two different GalNAc-conjugated siRNAs targeting vigilin (siVig-GalNAc#1: $n = 10$, siVig-GalNAc#2: $n = 10$) or PBS ($n = 9$) for 18 weeks starting at 4 weeks of age. Values are shown relative to PBS-injected control mice. (**b**) Immunoblot analysis of vigilin targets from blood plasma. ApoM and apoA-I were used as loading controls. Time course of plasma (**c**) cholesterol and (**d**) triglyceride levels throughout treatment period. Fractionated blood plasma from treated mice indicating VLDL, LDL and HDL particles that were quantified through (**e**) cholesterol and (**f**) triglyceride levels in each fraction. Quantification of (**g**) plasma NEFA, (**h**) hepatic triglyceride and (**i**) cholesterol levels. (**j**) Quantification of plasma bile acid levels. (**k,l**) Characterization of atherosclerosis in mice from **a**. (**k**) Representative oil red O-stained aortic root sections and (**l**) quantification of the lesion areas. Scale bar, 200 μm. *$P < 0.05$, **$P < 0.01$ and #$P < 0.05$, ##$P < 0.01$, ###$P < 0.001$ determined by ANOVA with Tukey's (in **c** and **d**) or Holm-Sidak (in **a**, **g**–**j**) post hoc analysis. All data are shown as the mean ± s.d.

obese, insulin-resistant subjects[51,52]. Given the high association between NAFLD/NASH and cardiovascular diseases[53,54], increased levels of vigilin in hepatic steatosis may therefore provide a potential link to increased risks for cardiovascular diseases.

## Methods

**Animal experiments.** All animal models shown were male and on a C57BL/6 N background and purchased from Janvier or Charles River. Mice used for isolation of primary hepatocytes were 8–10 weeks old. The ages of animals used for physiological experiments are indicated in the respective figure legends. Mice were housed in a pathogen-free animal facility at the Institute of Molecular Health Sciences at ETH Zurich. The animals were maintained in a temperature- and humidity controlled room on a 12 h light–dark cycle (lights on from 6:00 to 18:00). Mice were either fed a standard laboratory chow, a high-fat diet (for DIO mice; fat, carbohydrate, protein content was 45, 35 and 20 kcal%, respectively) (Research Diets, D12451) or chow diet AIN76 supplemented with 0.02% cholesterol[55] (for $Ldlr^{-/-}$ mice; Ssniff). All animal experiments were approved by the Kantonale Veterinäramt Zürich.

**Adenoviral infections.** The sequence of human V5-tagged VIGILIN was cloned into pVQAd CMV K-NpA (Viraquest) using the restriction sites BamHI and XhoI (NEB). Ad-GFP was based on the same vector backbone (including GFP) but lacked the insert transgene. shRNAs targeting vigilin (Supplementary Table 2) were cloned under a U6 promoter into the pVQAd AscI-NpA vector (Viraquest). Adenovirus production was performed at Viraquest, USA. All adenoviruses expressed *GFP* from an independent promoter. Adenoviral infection of mice was performed by a single tail vein injection of $3 \times 10^9$ plaque-forming units in a final volume of 0.2 ml diluted in PBS. Mice were sacrificed 7 (for gain-of-function experiments) or 10 days (for loss-of-function experiments) post injection.

**Primary hepatocytes isolation.** Mice were anaesthetized by intraperitoneal injection of 150 μl pentobarbital (Esconarkon US vet) pre-diluted 1:5 in PBS. The liver was perfused by cannulation of the hepatic portal vein with the caudal vena cava as a drain. The liver was perfused with pre- warmed Hank's Balanced Salt Solution (Life Technologies) containing 0.5 mM EGTA followed by pre-warmed digestion medium (DMEM 1 g l$^{-1}$ glucose (Life Technologies), 1% Penicillin–Streptomycin (Life Technologies), 15 mM HEPES (Life Technologies), 30 μg ml$^{-1}$ Liberase Research Grade medium Thermolysin concentration (Roche)) each for four minutes with a flow rate of 3 ml min$^{-1}$. The liver was surgically removed, hepatocytes released into 10 ml digestion media by shaking and supplemented with 15 ml ice-cold low glucose media (DMEM 1 g l$^{-1}$ glucose (Life Technologies), 1% Penicillin–Streptomycin (Life Technologies), 10% heat-inactivated fetal bovine serum (Sigma-Aldrich), 1% GlutaMax (Life Technologies)) and filtered through a 100 μm Cell Strainer (BD). The suspension was then washed three times with 25 ml of ice-cold low glucose media at 50g and 4 °C for 2 min. Hepatocytes were counted and plated at $4 \times 10^6$ cells surface-treated P10 plates (BD Primaria) in low glucose media. Three hours after plating, cells were washed once with PBS and medium was changed to Williams E medium (or methionine-free DMEM for cell extracts used for *in vitro* translation; Life Technologies) supplemented with 1% Penicillin–Streptomycin (Life Technologies), 1% GlutaMax (Life Technologies) and harvested 16 h (or 2 h for *in vitro* translation) after medium change[56]. All cells were incubated at 37 °C in a humidified atmosphere containing 5% CO$_2$.

**Blood plasma collection and measurements.** For measuring blood plasma insulin, ALT, triglyceride, cholesterol and NEFA levels, blood was collected from the submandibular vein in non-heparinized capillary tubes. EDTA was added to a final concentration of 5 mM as an anti-coagulant. Plasma was then separated by centrifugation at 8,000g for 4 min. Measurements were performed using commercial kits. Plasma insulin was measured with the Rat Insulin ELISA Kit (Crystalchem). Plasma cholesterol (Roche), triglycerides (Roche), NEFA (Wako) and bile acids (Crystalchem) were measured by colorimetric assays, according to the manufacturer's instructions.

***In vivo* VLDL secretion assay.** Gain-of-function and loss-of-function mouse models were fasted for 6 h and injected intravenously with the lipase inhibitor tyloxapol (500 mg kg$^{-1}$; Sigma) prior to blood collection at 0, 1, 2 and 3 h after injection. The collected blood samples were used for TG measurements and the VLDL-TG production rate was calculated from the slope of the plasma TG versus time curve.

**Metabolic labelling and immunoprecipitation.** Primary hepatocytes isolated from gain-of-function and loss-of-function mouse models were washed twice with PBS and incubated for 2 h in methionine/cysteine free medium. Cells were then pulsed for 2 h with 150 μCi of [$^{35}$S-]methionine/cysteine (Expre$^{35}$S$^{35}$S;

PerkinElmer) per well and chased for 8 h in full hepatocytes medium after washing with PBS. Metabolically labelled cells were harvested and lysed in 1 × NP40 lysis buffer (50 mM HEPES, pH 7.5, 150 mM KCl, 0.5 mM EDTA, 1 mM NaF, 0.5% (v/v) NP40, 50 μM DTT, complete EDTA-free protease inhibitor cocktail (Roche)) and incubated on ice for 10 min. Lysates were cleared by centrifugation at 13,000g before the supernatant was collected and used for immunoprecipitation. Vigilin targets were immunoprecipitated with target-specific antibodies conjugated to protein G Dynabeads (Life Technologies) over night at 4 °C. Ten microlitres of protein G magnetic particles were used per ml cell lysate. Magnetic beads were collected on a magnetic rack and washed with 1 × NP40 lysis buffer three times before quantification using the scintillation counter.

***In vivo* VLDL clearance assay.** Mice were injected intravenously with 100 μCi of [$^3$H]glycerol and blood was collected 1 h post injection. Plasma was extracted and fractionated to obtain the radiolabeled VLDL fraction. Mice under study from gain- and loss-of-function models were injected intravenously with 200,000 dpm of $^3$H-labelled VLDL. The disappearance of radiolabeled VLDL was determined from plasma samples drawn 30, 60, 90 and 120 min. after VLDL administration using scintillation counting[57].

**Antibodies.** The following antibodies were used in immunoblotting: rabbit anti-Hdlbp/vigilin (1:5,000) (Abcam, #ab109324), mouse anti-γ-tubulin (1:10,000) (Sigma-Aldrich, #T6557), rabbit anti-laminB (1:5,000) (Cell Signaling, #9087S), rabbit anti-Gapdh (1:500) (Santa Cruz, #2118S), rabbit anti-Rbm47 (1:5,000) (Abcam, #ab167164), rabbit anti-HuR (1:500) (Santa Cruz, #sc-20694), rabbit anti-Histone H3 (1:5,000) (Cell Signaling, #4499S), rabbit anti-apoB (1:2,000) (Meridian, #K23300R), rabbit anti-apoA-I (1:10,000) (Meridian, #K23500R), rabbit anti-fibronectin (1:5,000) (Abcam, #ab2413), goat anti-fetuin-A (1:500) (Santa Cruz, #sc-9668), rabbit anti-alpha1-Antitrypsin (1:1,000) (Proteintech, #16382-1-AP), rabbit anti-orosomucoid (1:1,000) (Proteintech, #16439-1-AP), rabbit anti-apoM (1:2,000) (self-made, #aa140-159), goat anti-albumin (1:10,000) (Bethyl, #A90-134A), mouse anti-V5 (1:5,000) (Invitrogen, #R960-25), rabbit anti-HA (1:5,000) (Abcam, #ab9110).

**PAR-CLIP in primary hepatocytes.** For PAR-CLIP, primary hepatocytes from Ad-VIGILIN injected mice (for overexpression of V5-tagged human VIGILIN) were isolated and supplemented with 100 μM 4SU for 16 h before crosslinking. After decanting the growth medium, living cells were irradiated with 0.15 J cm$^{-2}$ of 365 nm UV light. Cells were harvested and lysed in NP40 lysis buffer (50 mM HEPES, pH 7.5, 150 mM KCl, 0.5 mM EDTA, 1 mM NaF, 0.5% (v/v) NP40, 50 μM DTT, complete EDTA-free protease inhibitor cocktail (Roche)). The cleared cell lysates were treated with RNase T1 (at a final concentration of 1 U μl$^{-1}$ for 15 min at 22 °C). V5-tagged vigilin was immunoprecipitated with anti-V5 antibodies bound to 10 μl of Protein G magnetic particles per ml cell lysate prior to RNase T1 treatment (Fermentas) at a final concentration of 100 U μl$^{-1}$ for 15 min at 22 °C. Beads were washed and resuspended in dephosphorylation buffer (50 mM Tris-HCl (pH 7.9), 100 mM NaCl, 10 mM MgCl$_2$, 1 mM DTT). Calf intestinal alkaline phosphatase (CIP, NEB) was added to a final concentration of 0.5 U μl$^{-1}$ to dephosphorylate the RNA. To label the crosslinked RNA, beads were washed and incubated with [γ-$^{32}$P]ATP to a final concentration of 0.5 μCi μl$^{-1}$ and T4 PNK (NEB) to 1 U μl$^{-1}$ in one original bead volume for 30 min at 37 °C. Non-radioactive ATP was added to obtain a final concentration of 100 μM and incubated for another 5 min at 37 °C. The radiolabeled band corresponding to the 155 kDa VIGILIN–RNA complex was separated by SDS–PAGE and electroeluted. The electroeluate was proteinase K digested at a final concentration of 1.2 mg ml$^{-1}$ and incubated for 30 min at 55 °C. The RNA was recovered by acidic phenol/chloroform extraction and ethanol precipitation. The recovered RNA was turned into a cDNA library using 3′ and 5′ adapter ligations that were carried out on a 20 μl scale using 10.5 μl of the recovered RNA with chemically pre-adenylated adapter oligodeoxynucleotides (3′ adapter DNA, except for the riboadenylate rApp residue: 5′–3′ rAppTCGTATGCCGTCTTCTGCTTGT) and 5′ adapter RNA (5′–3′ GUUCAGAGUUCUACAGUCCGACGAUC) before reverse transcription and PCR amplification[58]. PCR products were cut out from a 3% NuSieve low-melting point agarose gel, the PCR product eluted from gel pieces using the GelElute kit (Qiagen) and sequenced at the Rockefeller University Genomics Center using the Solexa technology. Reads were adapter extracted with cutadapt, clipped with length of at least 20 nts and mapped to the mm10 mouse genome with Bowtie 0.12.9 (Bowtie parameters '-v 1 -m 10 –all –best –strata'), allowing for one mismatch. Processing and annotation of clusters to the ENCODE GRCm38 genome annotation was performed using the PARalyzer software with default settings as in Corcoran *et al.*[24] (http://www.genome.duke.edu/labs/ohler/research/PARalyzer/). Mapping statistics are given in Supplementary Data 1. The results from both PAR-CLIP replicates combined are listed in Supplementary Data 1 and 2. In addition, PAR-CLIP reads were mapped uniquely against the genome with 0 and 1 mismatches and annotated with bedtools using the ENCODE GRCm38 genome annotation to provide an independent measure of T-C transition, 1 mismatch and 0 mismatch reads per transcript (Supplementary Data 2). Targets were ranked by the sum of reads harbouring T-to-C conversions from both PAR-CLIP replicates.

**Motif analysis.** Kmer enrichment motif analysis was carried out calculating 4-mer enrichments by sliding within a 20-nt long window along PAR-CLIP clusters and using the shuffled (10,000 times) GRCm38 mouse protein-coding open reading frames as background sequences. Shuffled sequences were generated with the HMMER-3.0 suite. We further used the MEME suite, MEME (http://meme-suite.org/tools/meme) to define the motif of the top 759 protein-coding clusters (5′UTR, CDS, 3′ UTR) as defined by PARalyzer, which had at least 10 reads.

**RNA isolation and quantification.** RNA was extracted using Trizol (Life Technologies) according to the manufacturer's instructions, except for a 30 min isopropanol precipitation at $-20\,°C$. RNA integrity was analysed on an Agilent 2100 Bioanalyzer for all samples that were sequenced. RNA was subjected to DNase I treatment with the DNA-free kit (Invitrogen), when necessary. RNA was reverse transcribed using the High Capacity cDNA Reverse Transcription Kit (Applied Biosystems). Quantitative PCR was performed in an LC480 II Lightcycler (Roche) and using gene specific primers and Sybr Fast 2x Universal Master mix (Kapa). Results were normalized to 36B4 or Actb mRNA levels.

**Illumina RNA sequencing.** The quality of the isolated RNA was determined with a Qubit (1.0) Fluorometer (Life Technologies) and a Bioanalyzer 2100 (Agilent). Only those samples with a 260/280 nm ratio between 1.8 and 2.1 and a 28S/18S ratio within 1.5 and 2.0 were further processed. The TruSeq RNA Sample Prep Kit v2 (Illumina) was used for cDNA library preparation. Quality and quantity of the enriched libraries were validated using Qubit (1.0) Fluorometer and the Caliper GX LabChip GX (Caliper Life Sciences). Libraries were normalized to 10 nM and sequenced on the Illumina HiSeq 2000 at the Functional Genomics Center Zurich.

**Sequencing data analysis.** Reads were quality checked with FastQC. Reads at least 20 bases long, with a tail phred quality score greater than 20 were aligned to the reference genome and transcriptome (FASTA and GTF files, respectively, downloaded from the UCSC, genome build mm10) with STAR[56] with default settings for single end reads. Distribution of the reads across genomic isoform expression was quantified using the R package GenomicRanges[59] from Bioconductor Version 3.0. Differentially expressed genes were identified using the R package edgeR[60] from Bioconductor Version 3.0.

Most abundant liver transcript isoforms were matched to the Uniprot databank for signal peptides and transmembrane domains. The existence of signal peptides and transmembrane domains were handcurated using Protter[61].

**Western blot analysis.** Cells and tissues (using the Tissue Lyser II, Qiagen) were homogenized with 3 volumes of RIPA lysis buffer (50 mM Tris-HCl pH 8, 150 mM NaCl, 1% NP40, 0.5% sodium deoxycholate, 0.1% SDS and 1 tablet cOmplete EDTA-free protease inhibitor cocktail (Roche) per 50 ml buffer), incubated for 10 min on ice and centrifuged for 10 min at 20,000g and 4 °C. Protein concentrations were determined using the Bicinchoninic Acid Assay (Sigma-Aldrich). Equal protein amounts were boiled in Laemmli buffer (1.7% SDS, 5% glycerol, 0.002% bromophenol blue, 60 mM Tris-HCl pH 6.8, 100 mM DTT) for 5 min at 95 °C, separated by SDS–PAGE and transferred onto nitrocellulose membranes by electroblotting in a wet chamber (Bio-Rad). The membranes were blocked for 1 h with 5% non-fat dry milk TBS-0.1% Tween (Sigma-Aldrich), incubated with the primary antibodies overnight at 4 °C, followed by three washes in TBS-0.1% Tween and incubation with a horseradish peroxidase-conjugated secondary antibodies (Calbiochem) for 2–3 h. Blots were then developed by chemiluminescent detection with a Fujifilm analyzer (LAS-4000) and signals quantified using ImageJ. Uncropped scans of all of the immunoblots are shown in Supplementary Figs 7 and 8.

**Bacterial recombinant protein expression and purification.** Three litre cultures of *E. coli* BL21 (DE3)pLysS competent cells transformed with the pETM30-vigilin-His$_6$ construct were grown at 37 °C and 180 r.p.m. in Terrific Broth medium containing 75 μg ml$^{-1}$ Kanamycin until the OD$_{600}$ reached 1.3. The culture was then incubated at 18 °C for 1 h and protein expression was induced with 0.2 mM isopropyl-D-thiogalactopyranoside (IPTG). Incubation was then continued at 18 °C for 12 hrs. Cells were harvested by centrifugation for 10 min at 6,000g and 4 °C. All subsequent procedures were performed at 4 °C. Bacterial pellets were resuspended in chilled lysis buffer (50 mM Tris-HCl pH 7.5, 150 mM NaCl, 5 mM MgCl$_2$, 10% Glycerol, 1 mM β-Mercaptoethanol, 1 mM PMSF, 1 tablet EDTA-free protease inhibitor cocktail (Roche) per 50 ml, 1 mg ml$^{-1}$ Lysozyme (Sigma-Aldrich) 5 μg ml$^{-1}$ DNase I (Roche) in a ratio of 1 g cell-wet weight to 1 ml lysis buffer). The lysates were further sonicated in a pre-chilled 50 ml tube (Falcon) to reduce viscosity (5 s on, 20 s off, for 2 min, Amplitude: 28%), and insoluble material was removed by centrifugation for 30 min at 20,000g. The resulting supernatant was filtered through 0.45 μm polyethersulfone filter membranes (Filtropur S 0.45, Sarstedt). The lysate was diluted in 50 ml HisTrap-Buffer (50 mM Tris-HCl pH 7.5, 1 M NaCl, 5 mM MgCl$_2$, 10% Glycerol, 30 mM Imidazole pH 8.5), adjusted buffer pH to 7.6 with concentrated HCl before loading onto a 5 ml HisTrap HP column (GE Healthcare Life Science), pre-equilibrated in HisTrap-Buffer and attached to an ÄKTA Explorer FPLC. Diluted lysate was passed through the column at

2 ml min$^{-1}$ and then gradually eluted with increasing concentrations of HisTrap-Buffer containing 500 mM Imidazole, collecting 1 ml-sized fractions. The peak of fractions containing VIGILIN were determined by SDS–PAGE and Coomassie staining of the gel (typically at 150–200 mM imidazole). VIGILIN containing fractions were pooled, diluted in 50 ml Heparin-Buffer (50 mM Tris-HCl pH 7.6, 150 mM NaCl, 5 mM MgCl$_2$, 10% Glycerol, adjusted buffer pH to 7.6) and loaded on a 5 ml HiTrap Heparin HP column (GE Healthcare Life Science). RNA-depleted vigilin was eluted gradually using increasing concentrations of Heparin-Buffer containing 2 M NaCl and collected in 1 ml fractions. Eluate fractions were monitored by SDS–PAGE and Coomassie stainings. Fractions containing VIGILIN were pooled and dialyzed overnight using a 50 kDa MWCO Pur-A-Lyzer (Sigma-Aldrich) into storage buffer (20 mM Tris-HCl pH 7.6, 300 mM KCl, 5 mM MgCl$_2$, 50% glycerol, 1 mM DTT, 1 mM PMSF, 1 tablet EDTA-free protease inhibitor cocktail (Roche) per 50 ml). Aliquots of VIGILIN were stored at $-80\,°C$. Protein concentrations were determined by intensity of Coomassie staining in comparison to bovine serum albumin.

**Electrophoretic mobility shift assays.** Oligoribonucleotides were labelled with $[γ\text{-}^{32}P]$ATP and T4 polynucleotide kinase using standard conditions. A total of 10 nM $^{32}$P-labelled RNA was incubated with 0–10 μM protein in 20 μl reactions containing 250 mM KCl, 5 mM MgCl$_2$, 25 mM Tris-HCl pH 7.5, 10% glycerol, 1 mg ml$^{-1}$ acetylated BSA (Ambion), 1.5 μM of yeast tRNA (Invitrogen). Reactions were incubated at 25 °C for 5 min and separated on 1.2% agarose gel for 1 h at 130 V at room temperature using 1 × TBE. Agarose gels were dried under vacuum gel dryers at 60 °C for 2 to 3 h, exposed to a phosphoimager screen.

**Label-free mass spectrometry.** Medium from primary hepatocytes was collected 24 h after medium change and centrifuged at 14,000g to pellet insoluble remnants. Supernatants (60–80 μl) were precipitated with 1 volume of 20% TCA precipitation and washed twice with cold acetone. Dry pellets were dissolved in 45 μl buffer (10 mM Tris, 2 mM CaCl$_2$, pH 8.2) and trypsinized with 5 μl of 100 ng μl$^{-1}$ trypsin in 10 mM HCl for 30 min at 60 °C. Samples were dried, dissolved in 20 μl 0.1% formic acid and transferred to an autosampler vial for LC/MS/MS. Two microliter were injected. Label-free quantification of MS-data was performed by matching raw data to the Mouse Swiss-Prot database using MaxQuant[62]. Statistical analysis was then performed using Perseus (http://www.perseus-framework.org) after filtering out reverse hits, potential contaminants and entries identified by only one site. Protein intensities were normalized to the median sample intensity. Statistical significance was determined using a false discovery rate at 0.01, permutation-based multiple $t$-testing (250 ×) and a curve bend at s0 = 0.5.

**Plasma fractionation.** Lipoproteins from pooled plasma (200 μl total) were diluted in 1 mM EDTA–PBS and separated by FPLC using two Superose-6 FPLC columns in series (HR10/30) in 1 mM EDTA–PBS at 0.5 ml min$^{-1}$. Columns were calibrated using high and low molecular weight standards (GE Healthcare).

**Liver triglyceride and cholesterol content.** Lipids from 50 mg liver were extracted with 1 ml hexane:isopropanol (3:2) by homogenizing tissues using the TissueLyser II (Qiagen). Lysates were centrifuged at 20,000g for 3 min and the supernatant was transferred to a fresh tube. The pellet was re-extracted with 0.5 ml hexane:isopropanol, spun again and the supernatants were combined. A volume of 0.5 ml of 0.5 M Na$_2$SO$_4$ solution was added and the tubes mixed. The samples were centrifuged for 3 min at full speed and the upper organic phase was transferred to a fresh tube, avoiding contamination with the aqueous phase. The samples were spun again and the upper phase was transferred to a fresh tube and evaporated overnight under the fume hood. Lipids were dissolved in 1 ml of Triton X-100:methanol: butanol (1:1:3) mixture. Five microlitres were used for lipid quantifications.

**Oil red O stainings.** Frozen OCT-embedded liver pieces were stained with oil red O (ORO)[63]. Immediately after tissue collection livers were embedded in molds filled with OCT embedding matrix (CellPath) on dry ice and stored at $-80\,°C$. Tissues were cut with a cryostat into 10 μm thick sections. Sections were allowed to equilibrate for 10 min at room temperature. Sufficient ORO working solution (Abcam) was added to completely cover the sections and incubated for 10 min with the ORO solution. Slides were rinsed carefully in a stain dish under running tap water for 30 min. Slides were briefly dried and mounted with a water-soluble mounting medium and coverslips on them.

***In vitro* translation assay.** Full-length V5-tagged fetuin-A and apoM mRNAs were *in vitro* transcribed from pcDNA3.1 vectors using the mMESSAGE mMACHINE kit (Ambion). mRNAs were utilized for *in vitro* translation in 100 μl reactions using methionine- and cysteine-free amino acid mix and nuclease-treated extracts from primary hepatocytes isolated from mice injected with Ad-shCtrl or Ad-shVig[64]. Proteins were co-translationally radiolabeled by addition of 50 μCi [$^{35}$S]-methionine/cysteine (PerkinElmer) to the reaction. V5-tagged protein products were immunoprecipitated with a V5-antibody conjugated to protein G Dynabeads (Life Technologies), washed 10 × with IP wash buffer (50 mM

HEPES-KOH pH 7.5, 500 mM KCl, 0.05% NP40, 0.5 mM DTT, 1 tablet cOmplete EDTA-free protease inhibitor cocktail (Roche) per 50 ml) separated by SDS–PAGE and visualized by autoradiography using x-ray films (Fuji).

**Preparation of nuclear/cytoplasmic extracts.** Primary hepatocytes were permeabilized on ice in hypotonic lysis buffer (10 mM HEPES-KOH, pH 7.5, 1.5 mM MgCl$_2$, 10 mM KCl, 0.5 mM EDTA, 0.1% NP40, 1 mM DTT, 1 tablet cOmplete protease inhibitor cocktail) per 50 ml buffer (Roche) for 30 s, vortexed briefly and immediately centrifuged for 30 s at 8,000$g$ and 4 °C. After centrifugation, the supernatants (cytoplasmic extracts) were collected, and nuclear pellets were washed eight times in nuclear wash buffer (50 mM HEPES-KOH pH 7.5, 150 mM KCl, 2 mM EDTA, 0.5% NP40, 1 mM DTT, 1 tablet cOmplete protease inhibitor cocktail per 50 ml buffer (Roche)) by resuspension and centrifugation at 8,000$g$ for 30 s. Nuclear pellets were resuspended in RIPA buffer (50 mM Tris-HCl pH 8, 150 mM NaCl, 1% NP40, 0.5% sodium deoxycholate, 0.1% SDS and 1 tablet cOmplete EDTA-free protease inhibitor cocktail (Roche) per 50 ml buffer).

**Triglyceride secretion assay.** Seeded primary mouse hepatocytes extracted from mice injected with adenovirus were pulsed with 1 mM of pre-warmed albumin bound [1-$^{14}$C]-Palmitic Acid for 2 h and then washed three times with PBS. Williams E medium (Life Technologies) supplemented with 1% Penicillin–Streptomycin (Life Technologies), 1% GlutaMax (Life Technologies) was re-added to the cells and harvested 4 h after medium change. Incorporation of palmitic acid into triglycerides and subsequent secretion of radiolabeled triglycerides was quantified by extraction of the lipid fraction from the medium followed by liquid scintillation counting.

**GalNAc-conjugated siRNAs.** siRNA-GalNAc conjugates were synthesized as described previously[65]. The sequences and stabilizing nucleotide modifications of the siRNAs are shown in Supplementary Table 2. The GalNAc ligand was covalently linked to the 3′-end of the sense (S) strand of the siRNA by a phosphodiester linkage between the pyrrolidine scaffold and the S strand in all conjugate designs. A triantennary GalNAc ligand was synthesized to facilitate covalent conjugation to siRNAs[66]. Appropriately protected triantennary GalNAc monomer, which is compatible with solid-phase oligonucleotide synthesis and deprotection conditions, were synthesized using a *trans*-4-hydroxyprolinol moiety. The triantennary GalNAc building block was synthesized from D-( + )-galactosamine, 2-amino-2-(hydroxymethyl)-1,3-propanediol (TRIS), *trans*-4-hydroxy-L-proline methyl ester, and a glass solid support. The GalNAc monomer was then conjugated to sense strand of siRNA during solid-phase synthesis and deprotected to obtain the desired conjugate. Sense and antisense strands were synthesized on an ABI Synthesizer using commercially available 5′-O-(4,4′-dimethoxytrityl)-2′-deoxy-2′-fluoro-, and 5′-O-(4,4-dimethoxytrityl)-2-O-methyl-3-O-(2-cyanoethyl-N,N-diisopropyl) phosphoramidite monomers of uridine, 4-N-acetylcytidine, 6-N-benzoyladenosine, and 2-N-isobutyrylguanosine using standard solid-phase oligonucleotide synthesis and deprotection protocols. Phosphorothioate linkages were introduced by oxidation of phosphite utilizing 0.1 M DDTT in pyridine. After deprotection, ion-exchange HPLC purification followed by annealing of equimolar amounts of complementary sense and antisense strands provide the desired duplex sense and antisense strands were mixed and annealed by heating to 90 °C and slowly cooled to obtain the desired siRNAs. The siRNA samples were analysed by mass spectrometry, capillary gel electrophoresis and for endotoxin and osmolality, and the purity was ascertained.

**Quantification of atherosclerotic plaques.** Atherosclerosis was quantified in aortic root cross-sections from fresh-frozen optimal cutting temperature medium (OCT)-embedded hearts. Lesion size is shown as means of 5 frozen sections (10 μm), each 50 μm apart. Sections were stained with oil red O, hematoxylin and light green for visualization of atherosclerotic lesions[55]. Lesion areas were quantified using LAS software, version 4.2. (Leica). Analysis of the aortic roots was performed blindly without knowledge of the treatments.

**Statistical analysis.** Numerical values are reported as average ± s.d. unless stated otherwise. No statistical method was used to predetermine sample size, but sample size was based on preliminary data and previous publications as well as observed effect sizes. Outliers that were two standard deviations outside of the mean were routinely excluded from all analyses. Animals were sex- and age-matched. Animal studies were performed without blinding of the investigator. We assessed data for normal distribution and similar variance between groups using GraphPad Prism 6.0 if applicable. Some data sets had a statistical difference in the variation between groups. If not mentioned otherwise in the figure legend, statistical significance (*$P \le 0.05$, **$P \le 0.01$, ***$P \le 0.001$) was determined by unpaired two-tailed $t$-test, one-way ANOVA (when comparing $\ge$3 groups) or two-way ANOVA (for repeated measurements and time courses) with relevant post hoc tests (Holm-Sidak for = 3 groups and Tukey's for repeated time measurements and time courses). GraphPad Prism 6.0 software was used for statistical analysis of all data sets.

**Data availability.** The authors declare that the data supporting the findings of this study are available within the article and its Supplementary Information Files. RNA-Seq data were deposited at Gene Expression Omnibus under GSE78084. Supplementary Data 1 and 2 containing two tables listing PARalyzer identified RNA target sites and genome-wide RNA and proteomic experiments can be found with this article online at.

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

## Acknowledgements

We thank B. Bazylak-Kaps and the Functional Genomics Center Zurich for technical assistance. M.B.M. and S.G. were supported by a Boehringer Ingelheim Fonds PhD Fellowship. This work was supported from the European Research Council (ERC) grant 250210 'Metabolomirs' (M.S.), National Center of Competence in Research (NCCR) on RNA Biology and Disease (to M.S.), the Howard Hughes Medical Institute (HHMI) (T.T.), and by Swiss National Science Foundation grant 310030B _147089/1 (M.H.H.) and 310030_141209 (M.S.). Research in the laboratory of T.T. was supported by NIH transformative research award R01GM104962.

## Author contributions

M.B.M. and M.S. conceived the project; M.B.M. performed experiments, analysed and interpreted data and wrote the manuscript; S.G. performed bioinformatics analyses; D.T. analysed atherosclerotic lesions in aortic roots; M.M. and K.C. synthesized GalNAc-conjugated siRNAs; T.T. supervised PAR-CLIP and EMSA experiments. C.B. and M.H.H. recruited patients, collected clinical and blood chemistry data, and performed the liver biopsies. M.S. performed experiments, analysed and interpreted data, supervised the project and wrote the manuscript.

## Additional information

**Competing financial interests:** M.S. and T.T. are members of the Scientific Advisory Board and M.M. and K.C. are employees of Alnylam Pharmaceuticals. The other authors declare no competing financial interests.

