## [Peer Review File · Nature Communications]

Reviewers' comments:

Reviewer #1 (expert in atherosclerosis)

Remarks to the Author:

The study entitled "RNA-binding protein Vigilin is increased in hepatic steatosis and regulates VLDL secretion through modulation of Apob mRNA translation" by Mobin and colleagues identified Vigilin, a RNA-binding protein, upregulated in patients with NAFLD and in mouse models of obesity and insulin resistance. The authors provided convincing data demonstrating that Vigilin binds to the apoB mRNA regulating its translation. Overall, this study is of interest and could have important implications in the biology of lipoproteins. However, the authors have to provide additional information to support the major conclusion of the paper. There are also some issues with some of the experiments that need to be addressed.

Specific points:

1) It is surprising that the authors did not measure directly VLDL secretion in mice overexpressing Vigilin. This is an easy assay that is critical to support the author's conclusion. Similarly, the authors should measure VLDL secretion and clearance in mice treated with Vigilin inhibitors.

2) The authors should also measure clearance of VLDL particles and LPL activity. These are also a key experiments because Vigilin is expressed in the WAT and could influence LPL activity and VLDL lipolysis, thereby affecting circulating TAG and plasma VLDL levels.

3) The atherosclerosis studies are limited. Did the authors analyze atherosclerotic lesion using other methods (on face staining and carotid artery). How many mice were used in these experiments?. The lesions are also very small, it should be interesting to repeat the studies feeding mice a Western-type diet.

4) While the difference in plasma lipid levels between mice treated with GalNAc#1 and #2 is modest. GalNAc#2 does not attenuate atherogenesis (Is there a significant difference in plaque size in mice treated with PBS and GalNAc#2 ?)

In summary, this is an interesting study supported by high quality in vitro data. However, additional work in vivo is necessary to support author's conclusion.

Reviewer #2 (expert in lipid metabolism and atherosclerosis)

Remarks to the Author:

This manuscript describes interesting studies of the role of the RBP vigilin in hepatocytes. Using unbiased PAR-CLIP, vigilin was found to bind to a large number of secreted and transmembrane proteins. Using a loss of function approach primary hepatocytes, a few of these proteins were confirmed to be reduced in the media, consistent with a role of vigilin in promoting their secretion. Gain and loss of function studies in mice show effects on plasma levels of several proteins including apoB, apoC-III, fibronectin, and fetuin A. There are also physiological effects on plasma lipids, liver lipids (in some models), insulin sensitivity, and atherosclerosis. The most compelling LOF study was a chronic study performed using siRNA-GalNAc in LDLR KO mice. More data using this approach in other models would greatly enhance the manuscript.

Specific comments:

1. What was the effect of vigilin overexpression in ob/ob and DIO mice on hepatic fat and plasma lipids?

2. What is the effect of overexpression and knockdown of vigilin on VLDL TG and apoB production in vivo as assessed experimentally? These are critical experiments to prove the fundamental thesis that vigilin regulates hepatic apoB production. Studies of chronic KD using siRNA-GalNac would be the most compelling approach to address this question.

3. In siVig-GalNac treated animals, was VLDL TG and apoB production reduced? If the mechanism of the reduced plasma levels of apoB, VLDL and LDL is reduced VLDL production, then why aren't liver TG levels increased (as they were with short-term vigilin KD)?

4. The improvement in glucose tolerance and insulin sensitivity with siVig-GalNac treatment in the LDLR KO mice is interesting but mechanistically unexplained. The authors suggest that it may be due to reduction in fetuin A but this is not experimentally proven. Demonstration that this effect is abolished in fetuin A knockout mice or by simultaneous siRNA KD of fetuin A would directly implicate fetuin A in this process.

5. The effects of siVig-GalNac administration in a different model of insulin resistance and hepatic steatosis, such as ob/ob or DIO mice would link the chronic knockdown studies to the acute KD studies and provide additional information regarding LOF of vigilin in more classic insulin-resistant hepatic steatotic models.

6. In basal conditions, it is believed that the apoB protein is produced in excess with substantial ERAD degradation of non or poorly lipidated apoB. For a protein like vigilin that putatively increases apoB protein abundance to actually increase apoB secretion would require sufficient lipid availability to promote lipidation and secretion of apoB. Ideally the effects of vigilin KD in basal (ie chow-fed WT mice) and abundant lipid availability (ie DIO or ob/ob mice) would be compared, with the hypothesis that the effects on VLDL TG and apoB production and plasma TG and apoB levels would be much greater in the abundant lipid availability state. This would put these interesting findings in greater physiologic context.

7. The authors emphasize apoC-III as a target of vigilin but the data to support this are sparse. The reduced plasma levels of apoC-III could be secondary to the reduced apoB, TG, and VLDL rather than primarily caused by reduced apoC-III secretion. An experiment in primary hepatocytes or in vitro translation to show reduced apoC-III production would be of great interest, particularly given the interest in this protein as a therapeutic target.

Reviewer #3 (expert in RNA binding proteins, PAR-Clip)

Remarks to the Author:

Mobin and colleagues report high expression of the RNA-binding protein Vigilin in patients with the metabolic liver pathologies NAFLD and NASH, as well as in obese mice. Using viral vectors to elevate or reduce Vigilin abundance, the authors show that the levels of VLDL are positively regulated by Vigilin by controlling ApoB mRNA production. Global analysis of Vigilin target RNAs by PAR-CLIP analysis in primary hepatocytes led the authors to identify a CU-rich element with which Vigilin interacts, and additional targets (including Apob, Ahsg, Apoc3, Fn1 mRNAs) with phenotypic impact on cardiovascular function. The authors then demonstrate the potential therapeutic value of GalNac-conjugated Vigilin-directed siRNAs, which lowered VLDL/LDL levels and reduced atherosclerotic plaques in Ldlr^{-/-} mice.

Despite the abbreviated format, this report investigates many levels (biochemistry, molecular biology, cell biology, tissue, and system) of the role of Vigilin as a regulator of protein production in pathways relevant to liver lipid metabolism. The work is interesting, current, and potentially important therapeutically. Some issues that the authors need to address in order to complete and fully support their conclusions are below:

Major comments

1. The authors need to expand the description of the injections of viral vectors in mice. What mouse tissues overexpress Vigilin through this intervention? A survey of Vigilin expression in tissues is done for shVigilin, but needs to be shown for Vigilin overexpression as well. What makes the silencing of Vigilin so specific to liver (Fig. S1j)?
2. After Vigilin is overexpressed in mice, the authors must show the levels of Apob, Ahsg, Apoc3, Fn1 mRNAs and corresponding proteins (APOB, AHSG, APOC3, FN1).
3. The authors' PAR-CLIP analysis reveals a striking abundance of Vigilin binding sites in the coding region of target transcripts. With what pattern do Vigilin PAR-CLIP sites distribute along the CDS? Are they mostly clustered in 5' regions, along the translation start site? Towards the end (suggesting perhaps a role in translation termination)? Homogeneously along the mRNA? Near specific codons (perhaps rare codons)?
4. Specifically for the target mRNAs that the authors focus on (Apob, Ahsg, Apoc3, Fn1 mRNAs), where do the Vigilin PAR-CLIP sites lie? How are they distributed? A schematic showing this information would be helpful in order to visualize how Vigilin may be controlling translation.
5. Perhaps most importantly, the authors have not included critical experiments that demonstrate that in fact Vigilin promotes translation of target mRNAs. With the exception of analysis of translation in liver extracts, most of the authors' experiments support indirectly the notion that Vigilin promotes translation of the targets studied (Apob, Ahsg, Apoc3, Fn1 mRNAs). A number of standard analyses of translation must be included:
 - Side-by-side analysis of Apob, Ahsg, Apoc3, Fn1 mRNA levels and APOB, AHSG, APOC3, FN1 proteins in liver and hepatocytes after overexpressing and after silencing Vigilin are needed. Some of these data are already included in main or supplemental figures, but are not shown together, so reaching conclusions about protein changes in the absence of changes in mRNA is difficult.
 - Polysome distribution analyses for endogenous Apob, Ahsg, Apoc3, Fn1 mRNAs in hepatocytes/liver in which Vigilin is overexpressed or silenced will inform on whether Vigilin promotes translation initiation.
 - Translation assays in which nascent endogenous APOB, AHSG, APOC3, FN1 proteins are labeled metabolically are also necessary. The metabolic labeling of nascent translation from Apob, Ahsg, Apoc3, Fn1 mRNAs can be studied using the Click-iT® technology.

Minor comments:

- It is not clear if the authors excluded long noncoding RNAs from the PAR-CLIP analysis. If they did not, then authors could discuss this finding in the context of Vigilin's impact on translation.

Reviewer #4 (expert in atherosclerosis)

Remarks to the Author:

This is an interesting, original report that identifies ApoB mRNA, as well as the mRNAs of significant fraction of proteins secreted by hepatocytes, as a target of the RNA binding protein vigilin. The authors use a novel method called PAR-CLIP to identify the targets of Vigilin and show that these targets include several other secreted proteins that may influence atherosclerosis and insulin resistance and follow up with a Gal-Nac siRNA studying targeting Vigilin to show reduced atherosclerosis and improved insulin sensitivity. While the study is of great interest, there are some significant shortcomings.

1) The paper lacks a metabolic labeling study in primary hepatocytes to show directly that apoB synthesis and secretion are decreased by Vigilin knockdown and increased by Vigilin overexpression. Even though the apoB mRNA may be too long measure the translation of the full length mRNA, these metabolic labeling studies should be done in cells and possibly in vivo. The expected changes in apoB are not apparent in Fig 1m.

2) While the DIO diet is associated with increased hepatic Vilin expression and this correlates albeit weakly with plasma lipids and HOMA IR, there is no study of Vigilin knockdown in DIO mice to show reversal of these parameters. The correlations shown in Fig 1d are pretty meaningless without this kind of experiment, since they could simply indicate that both Vigilin and lipids/HOMA AR are increased by diet without establishing causation. It is possible that the relevant information can be gleaned from lower fat WTD experiments (Fig 4) assuming that it can be demonstrated that the WTD also increased Vigilin.

3) The data in Figs 2 and 3 is very compelling. The authors should consider moving Fig 1 to come after Figs 2 and 3, as it represents chasing a specific example, apoB.

4) Minor - what lipids contain the radioactivity in Fig 3F?

5) ApoC3 comes into the story later but is not apparent in Fig 3B. What is the interpretation of "other significant targets" that are increased in Fig 4b?

Point-to-point response to reviewer's comments

We would like to thank all reviewers for taking their time to carefully review our manuscript. We were delighted that all reviewers found our study interesting. We appreciate the constructive comments and have performed many additional experiments to further characterize the function of vigilin in hepatic metabolism and to support the conclusion of the study. Please find below the point-to-point response:

Reviewer #1 (expert in atherosclerosis)

Remarks to the Author:

The study entitled "RNA-binding protein Vigilin is increased in hepatic steatosis and regulates VLDL secretion through modulation of Apob mRNA translation" by Mobin and colleagues identified Vigilin, a RNA-binding protein, upregulated in patients with NAFLD and in mouse models of obesity and insulin resistance. The authors provided convincing data demonstrating that Vigilin binds to the apoB mRNA regulating its translation. Overall, this study is of interest and could have important implications in the biology of lipoproteins. However, the authors have to provide additional information to support the major conclusion of the paper. There are also some issues with some of the experiments that need to be addressed.

Specific points:

1) It is surprising that the authors did not measure directly VLDL secretion in mice overexpressing Vigilin. This is an easy assay that is critical to support the author's conclusion. Similarly, the authors should measure VLDL secretion and clearance in mice treated with Vigilin inhibitors.

We have measured VLDL secretion in mice following overexpression or silencing of vigilin in the liver using recombinant adenovirus. Increased VLDL secretion was measured upon overexpression and decreased VLDL secretion was observed when vigilin expression was reduced. These results support the major conclusion of our study and are shown as a new Fig. 5e and discussed on page 9 in the revised manuscript.

Following the reviewers suggestion we also measured the clearance of VLDL in mice treated with vigilin inhibitors. No changes in the clearance of

³H-labeled VLDL were observed compared to control mice. These results further support the conclusion of our study that changes in VLDL are due to vigilin-regulated VLDL synthesis/assembly rather than changes in VLDL clearance. The results are shown in a new Suppl. Fig. 5d and are mentioned on page 9.

2) The authors should also measure clearance of VLDL particles and LPL activity. These are also a key experiments because Vigilin is expressed in the WAT and could influences LPL activity and VLDL lipolysis, thereby affecting circulating TAG and plasma VLDL levels.

Clearance of VLDL particles were unaffected by vigilin (new Suppl. Fig. 5d). We also measured LPL activity and found it to be unaffected in mice treated with vigilin inhibitors or following vigilin overexpression compared to controls. These data are now shown in a new Suppl. Fig. 5c.

Please note that in the current study we are investigating the function of vigilin in the liver. All perturbations of vigilin are liver-specific. We therefore do not know if vigilin influences LPL activity, or other metabolic pathways, in other tissues.

3) The atherosclerosis studies are limited. Did the authors analyze atherosclerotic lesion using other methods (on face staining and carotid artery). How many mice were used in these experiments?. The lesion are also very small, it should be interesting to repeat the studies feeding mice a Western-type diet.

We agree that the atherosclerosis study is limited as we focused our investigations on the molecular and functional characterization of vigilin in the liver. Our observation that vigilin is a potent and positive regulator of apoB made the atheroprotective phenotype in mice, in which vigilin is silenced, predictable. This is due to the extensive knowledge of the role of apoB in atherosclerotic lesion development. We nevertheless believe that our atherosclerosis study is important, because it shows that reduced vigilin and apoB levels lead to the expected protection against atherosclerosis and that other vigilin targets do not oppose the predicted outcome. This is now discussed on page 9 (beginning of 2nd paragraph) and pages 11, 12.

As described in the figure legend, 10 mice were studied in each group. The quantification of the lesion area in oil red O stained aortic root sections

were carried out in a blinded fashion by 2 investigators. We chose Ldlr-KO mice of a C57BL/6 background due to the lipoprotein profile of these mice on a semisynthetic 0.02% cholesterol diet. These animals, in contrast to apoE-KO, develop a reasonable (pathophysiological) hypercholesterolemia of ≈ 500 mg/dL, which is 'LDL accentuated' and an accepted model for the assessment of atherosclerosis development (Teupser D, Persky AD, Breslow JL. *Arterioscler Thromb Vasc Biol.* 2003 23:1907-13.)

We did not perform an *en face* analysis to quantify the degree of atherosclerosis because this method can only be used if mice have a massive hypercholesterolemia (e.g. in apoE-KO or under prolonged treatment far beyond the 3 month treatment in our study in Ldlr-KO mice), which was not possible due to a limited GalNAc-siRNA availability.

Lastly, we agree that that it will be interesting to expand the atherosclerosis studies in other models, including Western-type diets. Of particular interest here is to investigate the relative contribution of various vigilin targets for the atheroprotective phenotype, i.e. whether the reduction of other vigilin targets that are potentially atheroprotective (i.e. apoC-III, hepatic derived fibronectin, fetuin-A), exert additive or even synergistic effects to apoB. However, such studies require simultaneous liver specific knockdowns, extensive optimization of chemically-modified siRNA sequences and GalNAc-conjugated siRNA chemistries, in addition to comprehensive atherosclerosis analysis in multiple models, which requires considerable resources and time and is beyond the current scope of this study.

4) While the difference in plasma lipid levels between mice treated with GalNAc#1 and #2 is modest. GalNAc#2 does not attenuate atherogenesis (Is there a significant difference in plaque size in mice treated with PBS and GalNA#2 ?)

Indeed, there is no significant difference in plaque size between PBS and GalNAc#2. In our study we tested two different GalNAc-siRNAs that exhibit different silencing activity. GalNAc#1 reduces hepatic vigilin levels by 80%, whereas GalNAc#2 only has a marginal, non-significant reduction compared to PBS injected mice (new Fig. 6a). Consequently, GalNAc#2 does not significantly affect plasma cholesterol, triglyceride, NEFA, VLDL levels and

atherosclerotic lesion size (new Fig. 6 c-f, k, l). In contrast, GalNAc#1-treated mice displayed highly significant reductions in cholesterol, triglyceride and VLDL levels. This explains why GalNAc#1 attenuates atherosclerosis, while GalNAc#2 does not. We included the inactive GalNAc#2 in our study as a further negative control (in addition to PBS injected mice), since this compound allows us to control for the effect of GalNAc-conjugated single strand oligonucleotides independent of siRNA activity. We are now making this point clearer in the text on bottom of page 9.

In summary, this is an interesting study supported by high quality in vitro data. However, additional work in vivo is necessary to support author's conclusion.

Reviewer #2 (expert in lipid metabolism and atherosclerosis)

Remarks to the Author:

This manuscript describes interesting studies of the role of the RBP vigilin in hepatocytes. Using unbiased PAR-CLIP, vigilin was found to bind to a large number of secreted and transmembrane proteins. Using a loss of function approach primary hepatocytes, a few of these proteins were confirmed to be reduced in the media, consistent with a role of vigilin in promoting their secretion. Gain and loss of function studies in mice show effects on plasma levels of several proteins including apoB, apoC-III, fibronectin, and fetuin A. There are also physiological effects on plasma lipids, liver lipids (in some models), insulin sensitivity, and atherosclerosis. The most compelling LOF study was a chronic study performed using siRNA-GalNAc in LDLR KO mice. More data using this approach in other models would greatly enhance the manuscript.

Specific comments:

1. What was the effect of vigilin overexpression in *ob/ob* and DIO mice on hepatic fat and plasma lipids?

We show that vigilin protein but not mRNA is approximately 2-3-fold increased in DIO and *ob/ob* mice compared to controls (new Fig. 1c and

Suppl. Fig 1d). To begin to understand the role of increased vigilin expression in these models we overexpressed vigilin in C57BL/6 wildtype mice using recombinant adenovirus.

Further overexpression of vigilin in obese mice might provide information if vigilin levels are limiting for apoB synthesis in obese animals. We therefore treated obese (*ob/ob*) mice with Ad-VIGILIN and measured hepatic triglycerides and plasma TG and cholesterol. However, we were (using a very high adenovirus titer) not able to further increase the levels of hepatic vigilin, indicating that expression of cellular vigilin is 'saturated' in models of obesity. Consequently, no changes in hepatic and plasma triglycerides were observed between Ad-VIGILIN and Ad-GFP treated mice (Suppl. Fig. 1 for reviewers only).

2. What is the effect of overexpression and knockdown of vigilin on VLDL TG and apoB production *in vivo* as assessed experimentally? These are critical experiments to prove the fundamental thesis that vigilin regulates hepatic apoB production. Studies of chronic KD using siRNA-GalNac would be the most compelling approach to address this question.

We have performed *in vivo* VLDL secretion studies in mice in which vigilin was either overexpressed or knocked down (see new Fig. 5e of revised manuscript). These new results confirm that modulation of vigilin also regulates VLDL/LDL secretion *in vivo*.

We also show in metabolic labeling experiments that apoB production is increased upon vigilin overexpression and reduced upon vigilin silencing in the liver (Fig. 5b, c). These data are now described on page 8 of the revised manuscript. As suggested, we also performed chronic knockdown of vigilin using siRNA-GalNac and show that VLDL secretion was suppressed in these mice (Suppl. Fig. 2 for reviewers only). Hence, these results further support the conclusion of our study that vigilin regulates hepatic VLDL secretion.

3. In siVig-GalNac treated animals, was VLDL TG and apoB production reduced? If the mechanism of the reduced plasma levels of apoB, VLDL and LDL is reduced VLDL production, then why aren't liver TG levels increased (as they were with short-term vigilin KD)?

To address if VLDL TG production was reduced in siVig-GalNac treated animals, we administered siRNA-GalNac to mice and employed the Triton WR-1339 method. siVig-GalNac#1 treated mice demonstrated significantly

reduced VLDL TG secretion when compared to the non-functional siVig-GalNAc#2 (Suppl. Fig. 2 for reviewers only). Furthermore, mice in which vigilin was either overexpressed or silenced, now shown in new Fig. 5e of the revised manuscript, show that VLDL TG production/secretion is increased upon overexpression of vigilin and decreased when vigilin expression is reduced. We also show that VLDL clearance is not significantly altered in mice in which hepatic vigilin was silenced (new Suppl. Fig. 5d).

Lastly, to provide more direct evidence that apoB production is directly influenced by vigilin we performed metabolic labeling experiments of apoB in primary hepatocytes of mice in which vigilin was either overexpressed or silenced (new Fig. 5b, c). These data (together with the unaltered apoB degradation and mRNA levels) provide direct evidence that vigilin regulates apoB at the level of translation.

We have consistently measured increased liver TG levels in the short-term vigilin knockdown study, a finding that we did not observe with the siVig-GalNAc knockdown (neither short term nor long term). We believe that the reason for this is that the adenovirus mediated vigilin knockdown is significantly stronger (>90%, new Fig. 2f) compared to the siVig-GalNAc#1 mediated knockdown, which is only \approx 75%. The differential efficiency of the vigilin knockdown is in turn reflected in the apoB levels. While the strong adenovirus mediated silencing of vigilin resulted in a 92% reduction of apoB, GalNAc#1 mediated knockdown reduced apoB levels by 72% (Suppl. Fig. 3 for reviewers only). Our experience with apoB knockdowns using antisense oligonucleotides or siRNAs support the interpretation that reducing hepatic apoB levels >90% leads to fatty liver whereas steatosis is not observed when apoB is only knocked down 70-80%.

4. The improvement in glucose tolerance and insulin sensitivity with siVig-GalNAc treatment in the LDLR KO mice is interesting but mechanistically unexplained. The authors suggest that it may be due to reduction in fetuin A but this is not experimentally proven. Demonstration that this effect is abolished in fetuin A knockout mice or by simultaneous siRNA KD of fetuin A would directly implicate fetuin A in this process.

We agree that the improvement in glucose tolerance and insulin sensitivity in the Ldlr-KO mice treated with siVig-GalNAc is mechanistically unexplained. The effect was small and was not observed in an extreme model of morbid obesity (see point 5 below). We have attempted to generate sh/siRNAs that will support >60% long-term silencing of fetuin-A in the liver, but were unable to identify sequences that exhibit the appropriate activities. Identification and careful validation of siRNAs for *in vivo* use require significant resources, and if done properly a ‘transcript walk’, to identify the appropriate siRNA followed by chemistry optimization to select for sequences with high activity and no associated compound/sequence related toxicities. This is especially important since it has been reported that fetuin-A signals through influencing TLR4-mediated inflammatory signaling. While we are committed to perform such studies in the future, we believe they are beyond the scope of our current study and, for the above reasons, are time consuming and require substantial resources.

In the revised manuscript we omitted the glucose tolerance and insulin sensitivity data obtained in Ldlr-KO mice, which remains unexplained. Our data show in multiple models that vigilin regulates lipid metabolism but has no effect on glucose metabolism: Neither vigilin overexpression in chow fed wildtype and *ob/ob* mice compared to respective controls, nor silencing of vigilin in wildtype chow fed and DIO as well as *ob/ob* mice affected glucose and insulin levels. These results demonstrate that vigilin’s primary action is in regulating lipid but not glucose metabolism.

5. The effects of siVig-GalNAc administration in a different model of insulin resistance and hepatic steatosis, such as *ob/ob* or DIO mice would link the chronic knockdown studies to the acute KD studies and provide additional information regarding LOF of vigilin in more classic insulin-resistant hepatic steatotic models.

As suggested by the reviewer we have studied the effects of vigilin knockdown using siVig-GalNAc in *ob/ob* mice. Animals were treated for 4 weeks. The effect of vigilin knockdown in this extreme obese model was less pronounced, with no improvement in plasma glucose levels and insulin sensitivity. However, we also measured improved plasma cholesterol levels and decreased VLDL and LDL levels (Suppl. Fig. 4 for reviewers only).

6. In basal conditions, it is believed that the apoB protein is produced in excess with substantial ERAD degradation of non or poorly lipidated apoB. For a protein like vigilin that putatively increases apoB protein abundance to actually increase apoB secretion would require sufficient lipid availability to promote lipidation and secretion of apoB. Ideally the effects of vigilin KD in basal (ie chow-fed WT mice) and abundant lipid availability (ie DIO or ob/ob mice) would be compared, with the hypothesis that the effects on VLDL TG and apoB production and plasma TG and apoB levels would be much greater in the abundant lipid availability state. This would put these interesting findings in greater physiologic context.

We compared the effects of vigilin silencing on VLDL and LDL TG levels between wildtype and DIO mice. Indeed, while a 90% knockdown of vigilin in wildtype (basal conditions) decreased VLDL and LDL TG levels by only 33%, an equal knockdown in DIO mice led to a decrease of 51% total VLDL/LDL TG (Suppl. Fig. 5 reviewers only). Hence, a vigilin knockdown under abundant lipid availability impacts VLDL/LDL levels in a more pronounced way than under basal conditions. This comparison therefore is in line the reviewer's notion that vigilin's biological role in VLDL/LDL secretion remains dependent on sufficient lipidation. These data are in good accordance with the current understanding of lipid secretion from the liver and support the view that lipid availability remains the limiting factor for hepatic VLDL/LDL and TG secretion. This finding is now discussed on page 11.

7. The authors emphasize apoC-III as a target of vigilin but the data to support this are sparse. The reduced plasma levels of apoC-III could be secondary to the reduced apoB, TG, and VLDL rather than primarily caused by reduced apoC-III secretion. An experiment in primary hepatocytes or in vitro translation to show reduced apoC-III production would be of great interest, particularly given the interest in this protein as a therapeutic target.

We agree that the reduced plasma levels of apoC-III could be secondary to the reduced apoB. To investigate this further we have performed metabolic labeling experiments in primary hepatocytes of mice in which vigilin was overexpressed or silenced. These data are now shown in the revised manuscript in new Fig. 5b, c and provide strong and direct evidence that

vigilin regulates production of the discussed targets, including apoC-III.

Reviewer #3 (expert in RNA binding proteins, PAR-Clip)

Remarks to the Author:

Mobin and colleagues report high expression of the RNA-binding protein Vigilin in patients with the metabolic liver pathologies NAFLD and NASH, as well as in obese mice. Using viral vectors to elevate or reduce Vigilin abundance, the authors show that the levels of VLDL are positively regulated by Vigilin by controlling ApoB mRNA production. Global analysis of Vigilin target RNAs by PAR-CLIP analysis in primary hepatocytes led the authors to identify a CU-rich element with which Vigilin interacts, and additional targets (including Apob, Ahsg, Apoc3, Fn1 mRNAs) with phenotypic impact on cardiovascular function. The authors then demonstrate the potential therapeutic value of GalNAc-conjugated Vigilin-directed siRNAs, which lowered VLDL/LDL levels and reduced atherosclerotic plaques in Ldlr^{-/-} mice.

Despite the abbreviated format, this report investigates many levels (biochemistry, molecular biology, cell biology, tissue, and system) of the role of Vigilin as a regulator of protein production in pathways relevant to liver lipid metabolism. The work is interesting, current, and potentially important therapeutically. Some issues that the authors need to address in order to complete and fully support their conclusions are below:

Major comments

1. The authors need to expand the description of the injections of viral vectors in mice. What mouse tissues overexpress Vigilin through this intervention? A survey of Vigilin expression in tissues is done for shVigilin, but needs to be shown for Vigilin overexpression as well. What makes the silencing of Vigilin so specific to liver (Fig. S1j)?

We have experience with this virus for over a decade (*Nature Medicine* 11:418-422, 2005; *Nature* 438:685-689, 2005; *Nature* 474:649-653, 2011; *Cell Metabolism* 17:436-447, 2013). The recombinant adenovirus substrain is based on (Ad) serotype 5 that has a remarkable specificity for the liver.

This has been tested in the past through extensive analysis of fluorescence microscopy of mice infected with this virus, which expresses GFP. We have also added a western blot analysis of a tissue panel from mice injected with either Ad-VIGILIN or Ad-GFP as control (new Suppl. Fig. 2) to document the liver-specific overexpression of vigilin.

2. After Vigilin is overexpressed in mice, the authors must show the levels of Apob, Ahsg, Apoc3, Fn1 mRNAs and corresponding proteins (APOB, AHSG, APOC3, FN1).

We are now showing the levels of Apob, Ahsg, Apoc3, Fn1 mRNAs and proteins following overexpression of vigilin in a revised Suppl. Fig. 4d, e and Fig. 4d, respectively.

3. The authors' PAR-CLIP analysis reveals a striking abundance of Vigilin binding sites in the coding region of target transcripts. With what pattern do Vigilin PAR-CLIP sites distribute along the CDS? Are they mostly clustered in 5' regions, along the translation start site? Towards the end (suggesting perhaps a role in translation termination)? Homogeneously along the mRNA? Near specific codons (perhaps rare codons)?

We agree that this is an important question and we looked at this very carefully. We find a homogenous distribution of binding sites across the coding region and only few clusters in the UTRs. We refer the reviewer to Fig. 3c and a new Suppl. Fig. 3a in which we show the T-to-C read cluster distribution of vigilin for mRNA targets in our PAR-CLIP experiment. Clusters are evenly distributed along the CDS in the transcripts and do not show specific enrichment at the start codon/5' regions or towards the end/ 3' region. Clusters in the 5'UTR and the 3'UTR were sparse. We also did not find any evidence of clustering near or at the translation initiation or termination site. Similarly, we were also unable to detect enrichment of PAR-CLIP sites at preferred codons or any other patterning, except for a continuous distribution along the CDS.

We have revised the manuscript and make this point clearer with a new spatial analysis of CDS exons (new Suppl. Fig. 3a), which is now mentioned in the text on page 5. The analysis reveals no significant enrichment along the CDS exons, indicating a uniform distribution across CDS exons.

4. Specifically for the target mRNAs that the authors focus on (Apob, Ahsg,

Apoc3, Fn1 mRNAs), where do the Vigilin PAR-CLIP sites lie? How are they distributed? A schematic showing this information would be helpful in order to visualize how Vigilin may be controlling translation.

We have now added a schematic illustration showing the vigilin PAR-CLIP sites of the targets discussed in the paper, including Apob, Ahsg, Apoc3, Fn1 mRNAs, as a close-up in a new Suppl. Fig. 4c.

5. Perhaps most importantly, the authors have not included critical experiments that demonstrate that in fact Vigilin promotes translation of target mRNAs. With the exception of analysis of translation in liver extracts, most of the authors' experiments support indirectly the notion that Vigilin promotes translation of the targets studied (Apob, Ahsg, Apoc3, Fn1 mRNAs). A number of standard analyses of translation must be included:

- Side-by-side analysis of Apob, Ahsg, Apoc3, Fn1 mRNA levels and APOB, AHSG, APOC3, FN1 proteins in liver and hepatocytes after overexpressing and after silencing Vigilin are needed. Some of these data are already included in main or supplemental figures, but are not shown together, so reaching conclusions about protein changes in the absence of changes in mRNA is difficult.
- Polysome distribution analyses for endogenous Apob, Ahsg, Apoc3, Fn1 mRNAs in hepatocytes/liver in which Vigilin is overexpressed or silenced will inform on whether Vigilin promotes translation initiation.
- Translation assays in which nascent endogenous APOB, AHSG, APOC3, FN1 proteins are labeled metabolically are also necessary. The metabolic labeling of nascent translation from Apob, Ahsg, Apoc3, Fn1 mRNAs can be studied using the Click-iT® technology.

We agree that it is important to provide more direct evidence demonstrating that vigilin promotes translation of target mRNAs. We are now showing side-by-side analyses of mRNA and protein levels of vigilin targets upon gain- and loss-of-function (Fig. 4d; Suppl. Fig. 4d, e). Following the reviewers suggestion, we also performed polysome distribution analyses for endogenous Apob, Ahsg, Apoc3, Fn1 mRNAs in primary hepatocytes, in which vigilin was overexpressed or silenced (see Suppl. Fig. 6 for reviewers only). We did not observe any changes in monosome, light polysome or heavy polysome levels for these targets upon overexpression or knockdown of vigilin. These data are therefore consistent with studies on

the yeast homologue SCP160 (Hirschmann et al.) and demonstrate that vigilin does not influence translation initiation.

Lastly and importantly, we directly show that vigilin promotes translation by metabolic ³⁵S-methionine/cysteine labeling experiments in primary hepatocytes in which vigilin was either overexpressed or silenced (see new Fig. 5b and c, respectively). These data provide additional and direct evidence that vigilin promotes translation of its target mRNAs and are now mentioned in the revised manuscript on page 8.

Minor comments:

- It is not clear if the authors excluded long noncoding RNAs from the PAR-CLIP analysis. If they did not, then authors could discuss this finding in the context of Vigilin's impact on translation.

We used the latest annotation for the mouse genome (GENCODE, GRCm38, mm10), which includes long noncoding RNAs. We refer the reviewer to Fig. 3b, which illustrates the distribution of the RNA targets in the PAR-CLIP experiment. As depicted, non-coding and especially long non-coding RNAs (under misc. RNA) were hardly identified as targets and had very low read counts. We conclude that long noncoding RNAs are not bound and do not seem to be regulated by vigilin, in agreement with a role of vigilin predominantly in translational regulation of protein-coding mRNAs.

Reviewer #4 (expert in atherosclerosis)

Remarks to the Author:

This is an interesting, original report that identifies ApoB mRNA, as well as the mRNAs of significant fraction of proteins secreted by hepatocytes, as a target of the RNA binding protein vigilin. The authors use a novel method called PAR-CLIP to identify the targets of Vigilin and show that these targets include several other secreted proteins that may influence atherosclerosis and insulin resistance and follow up with a Gal-Nac siRNA studying targeting Vigilin to show reduced atherosclerosis and improved insulin sensitivity. While the study is of great interest, there are some significant shortcomings.

1) The paper lacks a metabolic labeling study in primary hepatocytes to show directly that apoB synthesis and secretion are decreased by Vigilin knockdown and increased by Vigilin overexpression. Even though the apoB mRNA may be too long measure the translation of the full length mRNA, these metabolic labeling studies should be done in cells and possibly in vivo. The expected changes in apoB are not apparent in Fig 1m.

We agree that it is important to provide more direct evidence demonstrating that vigilin promotes translation of target mRNAs. We have performed metabolic ³⁵S-methionine/cysteine labeling experiments in primary hepatocytes, including apoB, in which vigilin was either overexpressed or silenced (see new Fig. 5b and c, respectively). These data (together with the unaltered apoB degradation and mRNA levels) provide additional and direct evidence that vigilin promotes translation of its target mRNAs and are now mentioned on page 8 of the revised manuscript.

2) While the DIO diet is associated with increased hepatic Vigilin expression and this correlates albeit weakly with plasma lipids and HOMA IR, there is no study of Vigilin knockdown in DIO mice to show reversal of these parameters. The correlations shown in Fig 1d are pretty meaningless without this kind of experiment, since they could simply indicate that both Vigilin and lipids/HOMA AR are increased by diet without establishing causation. It is possible that the relevant information can be gleaned from lower fat WTD experiments (Fig 4) assuming that it can be demonstrated that the WTD also increased Vigilin.

As suggested, we have performed an analysis of vigilin knockdown in DIO mice to compare vigilin levels with these parameters. While the correlation of vigilin to the plasma lipid levels was significantly reversed (see new Supplementary Fig. 2s), the HOMA-IR index did not reveal any correlation with hepatic vigilin levels. This is in agreement with a with studies in chow fed, diet induced and *ob/ob* mice in which we did not observe changes in blood glucose or insulin levels upon vigilin overexpression or silencing. To avoid a misleading message and clearly draw the focus of this paper on the lipid phenotype, we omitted the correlations between vigilin and HOMA-IR.

3) The data in Figs 2 and 3 is very compelling. The authors should consider

moving Fig 1 to come after Figs 2 and 3, as it represents chasing a specific example, apoB.

We have reorganized the figures to make them less dense but basically kept the order. Our reasoning for this is that we want to report the regulation of vigilin in disease models and humans as well as the metabolic phenotype observed upon overexpression and silencing first and then proceed to the consequences of this regulation through an unbiased functional and molecular characterization of vigilin. The study then follows up on apoB since it is the main target responsible for the 'VLDL phenotype'.

4) Minor - what lipids contain the radioactivity in Fig 3F?

This figure (new Fig. 5d) shows ^{14}C counts of radiolabeled palmitic acid incorporated into triglycerides of VLDL and secreted into the medium by primary hepatocytes upon overexpression and knockdown of vigilin.

5) ApoC3 comes into the story later but is not apparent in Fig 3B. What is the interpretation of "other significant targets" that are increased in Fig 4b? 'Other significant targets' refers to transcripts or corresponding proteins that were i) found to be bound by vigilin in the PAR-CLIP study (therefore referred to as "target", and ii) were significantly regulated (but not among the top 100 according to PAR-CLIP) in the secretome analysis of primary hepatocytes isolated from mice injected with either Ad-shCtrl or Ad-shVig. ApoC-III was not detected by mass spec analysis in the supernatants of these cells as abundantly as other proteins and therefore scores lower in the statistical analysis. Nevertheless, *ApoC3* mRNA was identified as a substrate of vigilin in our PAR-CLIP and regulation is as strong in western blot analysis as apoB100 (Suppl. Fig. 4b, Fig. 4d). We have now included apoC-III earlier in the manuscript and also added it to new Fig. 4d.

a

b

c

d

Supplementary Figure 1

Characterization of hepatic and plasma lipids in Ad-GFP and Ad-VIGILIN injected *ob/ob* mice.

(a) Western blot analysis of liver extracts from Ad-GFP and Ad-VIGILIN injected mice (1×10^{10} pfu/mouse). Each lane represents a different animal.

(b-d) Hepatic triglyceride levels (b), plasma triglyceride (c) and plasma cholesterol levels (d) in *ob/ob* mice injected with either Ad-GFP (Control) or Ad-VIGILIN. Mice were analyzed at day 7 postinjection, when adenoviral gene expression is at its maximum. Statistical significance was determined by student's t-test. (n = 6 for each group)

Supplementary Figure 2

Hepatic triglyceride levels and VLDL triglyceride secretion assay in GalNAc-siVig administered mice.

Mice were injected with 5mg/kg of active (#1, n = 6) and non-active (#2, n = 6) GalNAc-siRNAs every 5 days over a period of 2 weeks.

(a) Groups of GalNAc#1 and GalNAc#2 treated mice showed no difference in hepatic triglyceride levels after treatment.

(b) Animals were fasted for 3 hours before receiving an intravenous injection of 500 mg/kg tyloxapol to block lipases. Blood was collected at indicated time points and measured for plasma triglyceride accumulation. Mice that were treated with GalNAc#1 secreted significantly less VLDL triglycerides than GalNAc#2 injected mice.

*P < 0.05, ***P < 0.001, determined by two-way ANOVA with Holm-Sidak post hoc analysis.

All values are shown as the mean \pm s.d.

Supplementary Figure 3

Comparison of plasma apoB levels in Ad-shRNA and GalNAc-siRNA injected mice.

Quantification of apoB protein revealed a significant difference in the reduction of plasma levels between Ad-shVig and GalNAc-siRNA treated mice (n = 6 in each group). While a strong adenovirus-mediated silencing of vigilin (> 90%) resulted in a 92% reduction of apoB, GalNAc#1 mediated knockdown (≈80%) reduced apoB levels by 72%.

***P* < 0.01 as determined by student's *t*-testing. All values are relative protein levels from normalized to control conditions and are shown as the mean ± s.d.

Supplementary Figure 4

Metabolic characterization of GalNAc-siRNA treated *ob/ob* mice.

Obese (*ob/ob*) mice were treated with PBS (n = 4) or GalNAc-siRNAs (n = 8 for both #1 and #2) over a period of 4 weeks with subcutaneous injections at 5 mg/kg every 5 days.

(a) Western blot analysis of liver extracts from PBS, GalNAc#1 and GalNAc#2 injected mice.

(b) Quantification of knockdown revealed a weak ($\approx 46\%$), albeit significant knockdown of hepatic vigilin. No changes were determined in **(c)** an intraperitoneal insulin-tolerance test, **(d)** blood glucose, **(e)** body weight, **(f)** hepatic, and **(g)** plasma triglyceride levels. However, **(h)** total plasma cholesterol as well as **(i)** VLDL/LDL triglyceride and **(j)** VLDL/LDL cholesterol amounts were reduced upon knockdown of vigilin by GalNAc#1-siRNAs. Statistical significance was determined by ANOVA with Holm-Sidak **(b, f)** or Tukey's **(c-e, g, h)** *post hoc* analysis. Parameters in **d, e, g** and **h** were monitored over treatment period and are given from indicated time points after treatment start. All values are shown as the mean \pm s.d.

Supplementary Figure 5

Impact on total VLDL/LDL and apoB protein levels after silencing of hepatic vigilin in C57BL/6 mice in conditions of low and high lipid availability.

(a) Western blot analysis of apoB protein levels after an equal reduction of hepatic vigilin (90–95%) in both chow (low lipid availability; $n = 3$) and DIO mice (abundant lipid availability; $n = 3$).

(b) Quantification of apoB protein levels reveals a significantly higher impact on apoB (when compared to control conditions) in DIO mice than in chow fed mice. All values are shown as the mean \pm s.d. **: $P < 0.01$ as determined by student's t-testing.

(c) Reduction of VLDL and LDL amounts from fractionated blood plasma upon adenovirus mediated knockdown of vigilin in chow fed ($n = 6$) and DIO mice ($n = 6$). VLDL/LDL levels were reduced by 33% in chow fed mice as compared to 51% in DIO mice. Values are shown as the area under the curve from triglyceride levels of VLDL and LDL containing fractions.

Supplementary Figure 6

Polysome distribution analysis of vigilin mRNA targets under study.

(a) Polysome profiling of liver extracts from mice injected with either Ad-shCtrl (Control) or Ad-shVig. Liver extracts were prepared under polysome stabilizing conditions and fractionated on a 10–50% sucrose gradient. UV absorbance of fractions was measured at 254 nm to detect fractions of mRNAs in monosomes, light polysomes (2–3 ribosomes) and heavy polysomes (≥ 4 ribosomes).

(b-j) Quantitative PCR analysis of vigilin mRNA targets under study isolated from monosomes, light polysomes and heavy polysomes revealed no shift in their polysome distribution after hepatic vigilin knockdown. Two biological and four technical replicates were used per group. Please note in panel b mouse qPCR primers were used to measure vigilin mRNA. These primers do not detect the human transcript of Ad-VIGILIN.

No statistical significance was determined by student's *t*-test.

REVIEWERS' COMMENTS:

Reviewer #1 (Remarks to the Author):

The authors have performed additional experiments to support the major conclusion of the paper (Villilin regulation of VLDL production). I don't have further comments.

Reviewer #2 (Remarks to the Author):

None

Reviewer #3 (Remarks to the Author):

The authors have addressed my concerns adequately.

Reviewer #4 (Remarks to the Author):

The revised manuscript is improved and responsive to my previous critique.